# Interdependent action of KH domain proteins Krr1 and Dim2 drive the 40S platform assembly

Miriam Sturm[1], Jingdong Cheng[2], Jochen Baßler[1], Roland Beckmann[2] & Ed Hurt[1]

Ribosome biogenesis begins in the nucleolus with the formation of 90S pre-ribosomes, from which pre-40S and pre-60S particles arise that subsequently follow separate maturation pathways. Here, we show how structurally related assembly factors, the KH domain proteins Krr1 and Dim2, participate in ribosome assembly. Initially, Dim2 (Pno1) orchestrates an early step in small subunit biogenesis through its binding to a distinct region of the 90S pre-ribosome. This involves Utp1 of the UTP-B module, and Utp14, an activator of the DEAH-box helicase Dhr1 that catalyzes the removal of U3 snoRNP from the 90S. Following this dismantling reaction, the pre-40S subunit emerges, but Dim2 relocates to the pre-40S platform domain, previously occupied in the 90S by the other KH factor Krr1 through its interaction with Rps14 and the UTP-C module. Our findings show how the structurally related Krr1 and Dim2 can control stepwise ribosome assembly during the 90S-to-pre-40S subunit transition.

[1] Biochemistry Centre, University of Heidelberg, Heidelberg 69221, Germany. [2] Department of Biochemistry, Gene Center, Center for integrated Protein Science - Munich (CiPS-M), Ludwig-Maximillian University, Munich 81377, Germany. Correspondence and requests for materials should be addressed to E.H. (email: ed.hurt@bzh.uni-heidelberg.de)

The biogenesis of eukaryotic ribosomes is a complex and extremely energy-consuming process, during which actively growing cells devote most of their RNA polymerase I and II activities to the production of ribosomal RNA (rRNA) and the messenger RNAs encoding ribosomal proteins[1]. In order to produce functional ribosomes, ~200 assembly factors participate in this pathway by mediating folding, modification, and trimming of the pre-rRNA, coupled with incorporation of the ribosomal proteins themselves. Following these synthesis and first assembly steps, pre-ribosomal particles are restructured and compacted, processes during which they migrate from the nucleolus to the nucleoplasm, before export into the cytoplasm, where final maturation occurs[2–4].

In eukaryotes, ribosome biogenesis starts with the formation of a large precursor particle, called the 90S pre-ribosome or small subunit (SSU) processome[5,6], the three-dimensional (3D) structure of which has been recently solved by cryo-EM[7–9]. The 90S assembles co-transcriptionally around the 5′ end of the 35S pre-rRNA[5,6]. The 5′ external transcribed spacer (5′-ETS) recruits and organizes a number of modules termed UTP-A, UTP-B, UTP-C,

and U3 snoRNP, which, together with many other 90S factors, encapsulate the nascent rRNA, thereby stabilizing the first ribosome biogenesis intermediate[10–13]. The pre-rRNA embedded into this 90S particle undergoes extensive base modifications, folding and cleavage reactions at distinct sites that are guided by different small nucleolar RNAs (snoRNAs) and their associated assembly factors[5,14]. The box C/D U3 snoRNA is crucial to this process, because it base-pairs at multiple sites with the 35S pre-rRNA, both within the 5′-ETS and mature 18S rRNA[15,16]. Correct heteroduplex formation between U3 and pre-rRNA is prerequisite for the early cleavage events to occur at sites $A_0$ and $A_1$ that yield the mature 5′ end of the 18S rRNA[17]. Eventually, the DEAH-box helicase Dhr1 and its activator Utp14 contribute to the dissociation of U3 from the 90S particle, which allows formation of an rRNA pseudoknot secondary structure at the decoding center of the small 40S subunit[18,19]. Following pseudoknot formation, a final cleavage occurs at site $A_2$, which marks the separation of the pre-40S and pre-60S maturation pathways[20,21]. While the pre-60S particles undergo a series of additional processing, maturation, and checkpoint steps in the nucleus before export into the cytoplasm[2], the pre-40S subunit emerges following the removal of the remaining 90S factors, before it rapidly leaves the nucleus with only a handful of biogenesis factors attached[22]. In the cytoplasm, final maturation occurs, which requires structural rearrangements at the head region of the pre-40S particle[23] and cleavage of the 20S pre-rRNA at site D by the endonuclease Nob1 to generate the mature 3′ end of the 18S rRNA[24–26]. This last processing event is stimulated by the initiation factor eIF5B and mature 60S subunits, which mimic a translation-like cycle as a final proofreading step for correct 40S biogenesis[27].

Dim2 and Krr1 are structurally related ribosome assembly factors, which belong to the family of RNA-binding proteins containing KH domains. Dim2 and Krr1 harbor two conserved KH motifs in sequence (KH1, KH2), but with different N- and C-terminal extensions (for sequence alignment, see Supplementary Fig. 1). Notably, the KH1 domains in both Krr1 and Dim2 lack the typical GXXG RNA-binding motif and instead participate in protein–protein interactions[28,29]. For example, Krr1 binds via its KH1 (KH-like) domain to Kri1, a nucleolar assembly factor associated with snR30[30], whereas the KH1 domain of Dim2 provides a binding site for the endonuclease Nob1[28]. In the case of Krr1, KH2 contacts RNA but it can also bind to the ribosome biogenesis factor Faf1[29]. To date, Dim2 has been predominantly studied in the context of the late pre-40S maturation pathway though its association with Nob1[25,28], although a role in early ribosome assembly and $A_2$ cleavage has been reported[31]. In contrast, Krr1 is exclusively associated with 90S particles and involved in 35S pre-rRNA processing[6,30].

In this study, we investigate how Krr1 and Dim2 participate during the early steps of ribosome biogenesis. Our data reveal the roles of these two KH domain proteins during the maturation of the 90S into pre-40S particles. We identify a novel link between Dim2 and the Dhr1 helicase and its co-factor Utp14, which together dismantle the U3 snoRNP from the 90S pre-ribosome. Moreover, we find that Krr1 interacts with ribosomal proteins Rps14 (uS11) and Rps1 (eS1) at the undeveloped 40S platform site of the 90S pre-ribosome, at which Krr1 also mediates recruitment of the UTP-C module. Thus, Krr1 occupies a site on the 90S particle, where later Dim2 associates after Krr1 removal, which is adjacent to Rps14 at the platform/head area of pre-40S subunit.

## Results

**Dual role of KH domain factor Dim2 in ribosome biogenesis.** To assess the in vivo role of the conserved KH-domain factor Dim2, we depleted Dim2 from yeast cells by repression of GAL::DIM2 expression upon shift from galactose- to glucose-containing medium for 8 h. Subsequently, we affinity-purified another assembly factor, Rio2[32], which is only co-enriched on pre-40S particles, and Enp1, which is present on both 90S and pre-40S particles[22]. Rio2–FTpA (Flag–TEV–ProtA) particles affinity-purified from non-depleted cells did not show a significant difference in co-enriched bands compared to those from Dim2-depleted cells, but the endonuclease Nob1 was considerably absent from this pre-40S particle under conditions of Dim2 depletion (Fig. 1a, western; note that the prominent Rio2 band co-migrates with the Nob1 band on the Coomassie-stained gel; for mock control of tandem affinity-purification, see Supplementary Fig. 2a). This finding is consistent with a previous study, which showed that Nob1 was dissociated from pre-ribosomal particles and found as soluble protein on the top of a sucrose gradient upon Dim2 depletion[28].

When Enp1 was affinity-purified from Dim2-depleted cells, the typical profile of the Enp1 co-enriched bands, which are mainly late pre-40S factors (e.g., Rrp12, Tsr1, Nob1, Dim1, Dim2), changed in favor of a massive co-enrichment of normally under-represented early 90S factors including UTP-A, UTP-B, UTB-C, and U3 snoRNP subunits (Fig. 1a). This finding underscores Dim2's essential role in 90S ribosome biogenesis. These different pre-ribosomal intermediates formed during Dim2 depletion were further characterized by sucrose gradient centrifugation (Supplementary Fig. 2b). As anticipated, Enp1-FTpA purified from Dim2-expressing cells recovered both pre-40S particles (lanes 6 and 7) and 90S pre-ribosomes (lane 10). In contrast, Enp1-FTpA purified from Dim2-depleted cells yielded an increased amount of 90S particles (lane 10) and an additional pool of abnormal pre-40S particles, which was enriched not only in late 40S factors (e.g., Tsr1, Rrp12), but also in a number of 90S factors (e.g., Rrp5, Utp10, Kre33, Krr1). Notably, this atypical pre-40S pool

**Fig. 1** Dim2 depletion blocks 90S biogenesis and causes a defect in Nob1 binding. **a** Yeast strain GAL::scDIM2 with integrated scEnp1-FTpA or scRio2-FTpA was grown in galactose (GAL) medium or shifted for 8 h to glucose (GLU)-containing medium before the indicated bait proteins were tandem affinity-purified in two consecutive steps involving the ProtA- in the first and the Flag-tag in the second step. TCA-precipitated final Flag eluates were analyzed by SDS-PAGE (4–12%) and Coomassie staining (upper panel) or western blotting (lower panel) using the indicated antibodies. The indicated major protein bands were identified by mass spectrometry. Note that Dim1 co-migrates with HA-Dim2. **b** Recombinant MBP (maltose-binding protein)-ctDim2 co-expressed with $HIS_6$-ctNob1 in E. coli BL21 cells was affinity-purified utilizing SP-Sepharose and subsequently Ni-NTA resin. The final eluate was fractionated by size-exclusion chromatography (SEC). Relevant SEC fractions (lanes 15–21) containing the heterodimer were analyzed by SDS-PAGE (4–12%) and Coomassie staining. **c** Schematic drawing of the different ctNob1 domains. The middle (MID) domain of Nob1 contains a highly conserved tryptophan at position W267, which was mutated and tested by binding to Dim2. **d** In vitro binding assay with immobilized GST-ctDim2 as bait and indicated $HIS_6$-ctNob1 prey constructs (input, lanes 1–5). BL21 extract was used as mock control (input, lane 6). Samples (lanes 7–12) are SDS eluates analyzed by SDS-PAGE and Coomassie staining. **e** Single-point mutation in the ctNob1-MID sequence causes loss of interaction with ctDim2. Recombinant $HIS_6$-ctDim2 co-expressed with either ctNob1 or ctNob1 W267G in E. coli BL21 cells were tandem-purified on SP-Sepharose and Ni-NTA beads. Input Ni-NTA, lanes 1 and 3. Final eluates (lanes 2 and 4), analyzed by SDS-PAGE (4–12%) and Coomassie staining. The experiments were performed at least twice with consistent results. Uncropped images are shown in Supplementary Fig. 9. S molecular weight protein standard, L load

contained nuclear exosome factors (e.g., Dis3) (lane 6), suggesting that the RNA in these intermediates is a target for exonucleolytic degradation, as previously shown for the 5′ ETS particle[33] (Supplementary Fig. 2c). Consistent with these findings, western blot analysis revealed reduced Nob1 levels in comparison to the other pre-40S assembly factors Tsr1 and Rio2 (Supplementary Fig. 2d), indicating that a Dim2-mediated maturation step is required for Nob1 recruitment.

To gain insight into the mechanism of Dim2-Nob1 interaction, we performed binding studies using thermostable Dim2 and Nob1 from *Chaetomium thermophilum*[34]. *ct*Dim2 and *ct*Nob1 readily formed a complex in vitro, which eluted as a heterodimer during size-exclusion chromatography (Fig. 1b). To characterize the interaction domain in greater detail, we dissected Nob1 in five domains (Fig. 1c), and performed in vitro binding studies with various *ct*Nob1 truncation constructs utilizing immobilized GST-*ct*Dim2. We observed that deletion of the PIN domain weakened but did not completely abolish binding to Dim2, whereas deletion of the middle (MID) domain blocked the Nob1-Dim2 interaction (Fig. 1d). Consistent with this finding, mutation of a highly

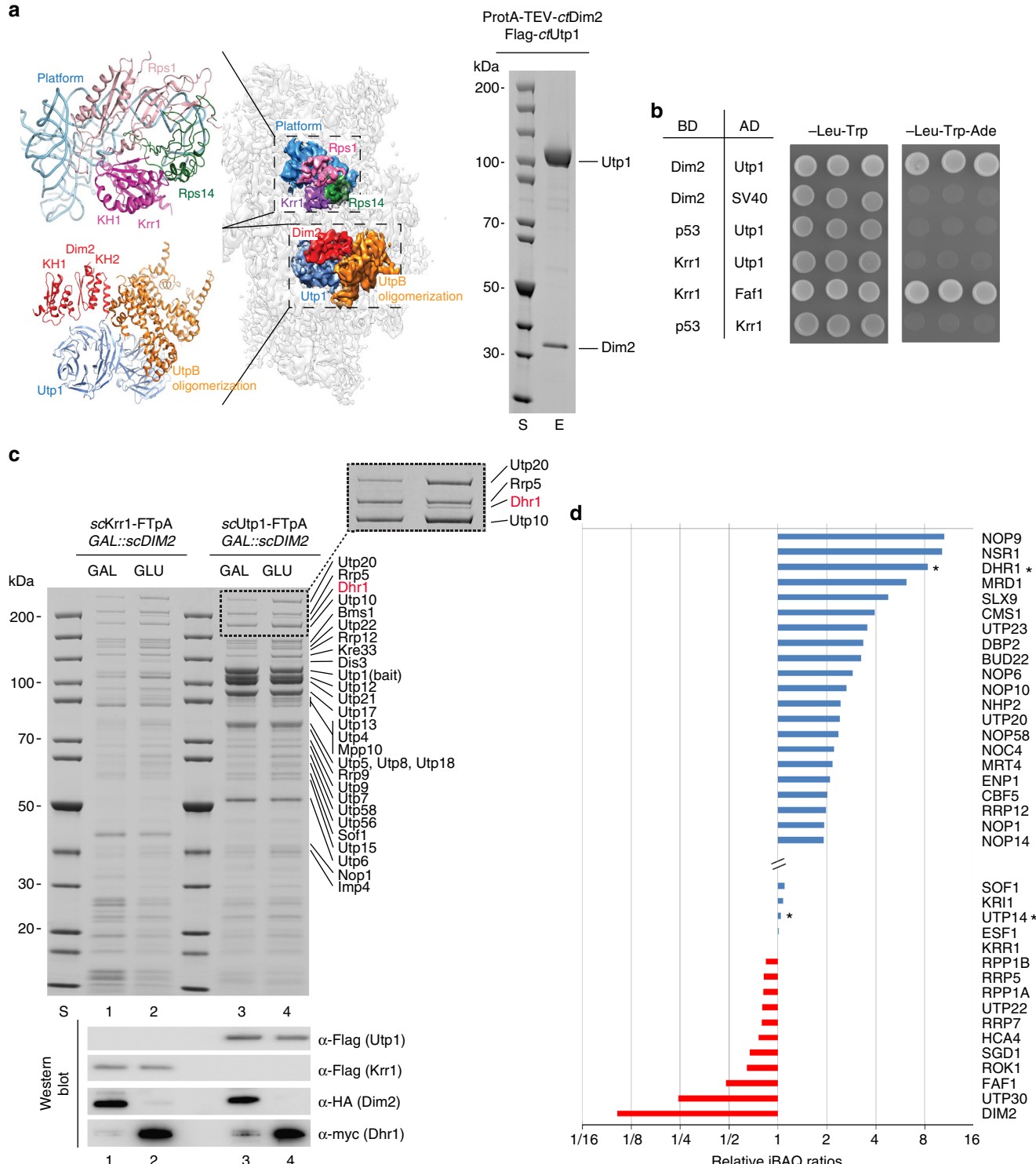

conserved tryptophan in the *ct*Nob1 middle domain (W267; Fig. 1c) inhibited *ct*Nob1-Dim2 heterodimer formation (Fig. 1e). Similarly, when the conserved MID motif was deleted from the yeast Nob1, Dim2 binding was abolished (Supplementary Fig. 3a). However, the sequence between PIN and MID domains (called Flex; Fig. 1c) was not required for Dim2 interaction. Accordingly, co-expression of the separate Nob1-PIN (1–169) domain in trans with a construct composed of MID-ZnF (250–354) domain allowed complex formation with Dim2 (Supplementary Fig. 3b).

To assess the in vivo role of Nob1-Dim2 complex formation, we deleted the MID domain from yeast Nob1 and tested for complementation of the otherwise non-viable yeast *nob1* null mutant. However, yeast cells expressing Nob1ΔMID did not exhibit a growth defect (Supplementary Fig. 3c) and association of Nob1ΔMID with late pre-40S particles, affinity-purified via the Ltv1 bait, remained unaffected (Supplementary Fig. 3d). Again, this finding is in line with the study by Woolls et al.[28], revealing that specific Dim2 point mutations, which impair Nob1 binding in vitro, did not significantly disturb the association of Nob1 with pre-ribosomal particles in vivo.

Taken all data together, the in vitro interaction between Nob1 and Dim2, which is mainly dependent on the Nob1 MID domain, plays a minor role in vivo, suggesting that targeting of these two assembly factors to the pre-40S particles follows redundant mechanisms.

**Dim2 interacts with UTP-B in the 90S pre-ribosome**. To better define the role of Dim2 in early 90S biogenesis, we looked for its binding partners on the 90S pre-particle. Previously, we have mapped two densities on the cryo-EM structure of the *ct*90S pre-ribosome to KH domain-folded proteins[7] (Fig. 2a). Mass spectrometry (MS) indicated that only two KH proteins, *ct*Dim2 and *ct*Krr1, were present on the purified *ct*90S particle, but due to the limited resolution, it was not possible to precisely assign them. Meanwhile, Dim2 and Krr1 could be localized in the yeast 90S structure[9]. Based on our density map of the thermophile 90S particle, both KH-like densities locate at the backpack of the pre-90S particle, but one of the two KH proteins is in direct contact with the UTP-B factor Utp1, whereas the other contacts the 90S-associated ribosomal proteins Rps14 and Rps1, which are located at the platform region of the mature 40S subunit but are already present on the 90S particle[35]. Dim2 and its neighbor Nob1 have been previously mapped to this exact platform region of the late pre-40S particle[36]. Thus, we rationalized that Dim2 could be recruited to the 90S particle at the Rps14–Rps1 site, where it remains associated until the pre-40S particle has formed. However, other data were not consistent with this view, as a two-hybrid interaction between *ct*Dim2 and *ct*Utp1 (Pwp2), a subunit of the UTP-B module, was observed[37], which is ~65 Å distant from the KH-like density associated with Rps14 and Rps1 on the 90S pre-ribosome (Fig. 2a, left panel).

Hence, we performed biochemical tests to determine how Dim2 and Krr1 physically interact with other factors on the 90S pre-ribosome. First, we selected *ct*Dim2 to identify stable binding partners. We could readily reconstitute a stoichiometric complex between *ct*Dim2 and *ct*Utp1 by co-expression and subsequent affinity-purification from yeast (Fig. 2a, right panel), consistent with our previous two-hybrid data[37]. In contrast, the structurally related KH factor *ct*Krr1 was not observed to interact with *ct*Utp1 in two-hybrid experiments, but showed a strong two-hybrid interaction with its known binding partner Faf1[29] (Fig. 2b). To determine which domain of Dim2 interacts with Utp1, we performed additional two-hybrid tests using *ct*Dim2 as the bait, which was divided into an N-terminal domain (residues 1–78), a KH-like domain (79–163; known to interact with Nob1), and a C-terminal KH domain (164–260; known to bind RNA[31]; Supplementary Fig. 4a). This analysis showed that *ct*Dim2 utilizes its KH-like domain (KH1) to interact with *ct*Utp1, which is reminiscent of how Dim2–KH1 binds to Nob1[28]. Thus, our data revealed that within the thermophile 90S particle, *ct*Dim2 is strategically positioned at the UTP-B module through its interaction with Utp1, suggesting that the other KH-like density adjacent to Rps14–Rps1 represents *ct*Krr1 (see below).

**Dim2 depletion causes trapping of Dhr1 on the 90S pre-ribosome**. To investigate the significance of the Dim2–Utp1 interaction in yeast, we performed affinity purifications of the 90S factor Krr1, in addition to Utp1 itself, in the presence and absence of Dim2. The pattern of co-isolated 90S assembly factors, stained by Coomassie, did not significantly change, except for a single band of ~180 kDa identified as Dhr1, which co-enriched in both Krr1- and Utp1-derived 90S particles in the absence of Dim2 (Fig. 2c). Semi-quantitative mass spectrometry (MS) of the Krr1–FTpA purification showed that Dhr1, a helicase essential for removing U3 snoRNA from the 90S pre-ribosome[18,19], was eightfold co-enriched if Dim2 was depleted (Fig. 2d). Interestingly, levels of its co-factor Utp14 remained unaffected. Moreover, some other assembly factors such as Nop9, Nsr1 and Mrd1 were also co-enriched during Dim2 repression, but this was not further followed in this study. Instead, we focused on Dhr1, since Utp14 (an activator of Dhr1) was recently found to interact with Dim2 in a 2-hybrid screen[37]. To investigate the link between Dim2 and Dhr1, we performed two-hybrid tests, but could not observe a direct interaction (Fig. 3a). However, and also

**Fig. 2** Interaction of Dim2 with biogenesis factors of the 90S pre-ribosome. **a** *ct*Dim2 specifically interacts with Utp1 (Pwp2) on the 90S pre-ribosome. Left panel: cryo-EM structure of the *ct*90S pre-ribosome (PDB: 5JPQ)[7] highlighting the position of Krr1 (violet) and interacting proteins Rps14 (green) and Rps1 (pink) in the context of the 40S platform pre-rRNA, (light blue), as well as Dim2 (red) and interacting Utp1 (dark blue) in the context of the UTP-B oligomerization platform (orange). Right panel: Flag₃-*ct*Utp1 and ProtA-TEV-*ct*Dim2 were co-expressed in yeast and tandem affinity-purified via the split-tag method. The final eluate (E) was analyzed by SDS-PAGE (4–12%) and Coomassie staining. **b** Dim2 but not structurally related Krr1 interacts with Utp1 by 2-hybrid. Yeast 2-hybrid analysis of three individual transformants, harboring the indicated bait (BD) and prey (AD) plasmids. Transformants were analyzed by growth for 2 days at 30 °C on SDC-Leu-Trp (plating efficiency) and SDC-Leu-Trp-Ade plates (strong interaction). SV40 and p53 served as negative controls. **c** Yeast strain *GAL::scDIM2* with integrated *sc*Krr1-FTpA or *sc*Utp1-FTpA was grown in galactose (GAL) medium or shifted for 8 h to glucose (GLU) containing medium before the indicated bait proteins were tandem affinity-purified. TCA-precipitated Flag eluates were analyzed by SDS-PAGE (4–12%) and Coomassie staining (upper panel) or western blotting (lower panel) using the indicated antibodies. The indicated proteins were identified by mass spectrometry. Dashed box is a zoom of the respective gel region to better display the Dhr1 enrichment. **d** Semiquantitative mass spectrometry of the Krr1–FTpA eluates from *GAL::scDIM2*-expressing (GAL) and *GAL::scDIM2* (GLU)-depleted cells (see Fig. 2c, lanes 1 and 2). The iBAQ (intensity-based absolute quantification) numbers derived from mass spectrometry analysis of the two different Krr1–FTpA eluates were normalized to the iBAQ value of the Krr1 bait protein. The relative iBAQ ratios of co-purified 90S factors derived from depleted versus expressing Dim2 cells are displayed, with the numerical data shown in Supplementary Data 1. The experiments were performed at least twice with consistent results. Uncropped images are shown in Supplementary Fig. 9. S molecular weight protein standard

consistent with our previous data[37], we observed a robust two-hybrid interaction between *ct*Utp14 and *ct*Dim2 (i.e., growth on an SDC-Trp-Ade plate), in which Dim2 utilized its KH-like domain to bind to Utp14 (Fig. 3a and Supplementary Fig. 5a, b). In contrast, the structurally related Krr1 did not show such a strong two-hybrid interaction with Utp14 (Supplementary Fig. 5a).

Based on these 2-hybrid data, we performed biochemical reconstitution with recombinant proteins. This allowed us to assemble a stoichiometric *ct*Utp14–Dim2 heterodimer through co-expression of the *C. thermophilum* orthologs in yeast, followed

by split-tag affinity-purification (Fig. 3b, left panel). As we were able to successfully reconstitute a *ct*Utp14–Dim2–Utp1 trimer, we conclude that Utp14 and Utp1 can simultaneously bind to the KH-like domain of Dim2 (Fig. 3b, middle panel). Analogously, we could show that Utp14 can act as bridging factor between Dim2 and Dhr1 (Fig. 3b, right panel). Accordingly, Dim2 might influence Dhr1 activity during the 90S-to-pre-40S transition (Discussion).

To show that the Dim2-Utp14 interaction is important in vivo, we performed genetic studies in yeast. For this purpose, we used a point mutation in the KH-like domain of Dim2 (W113D), which

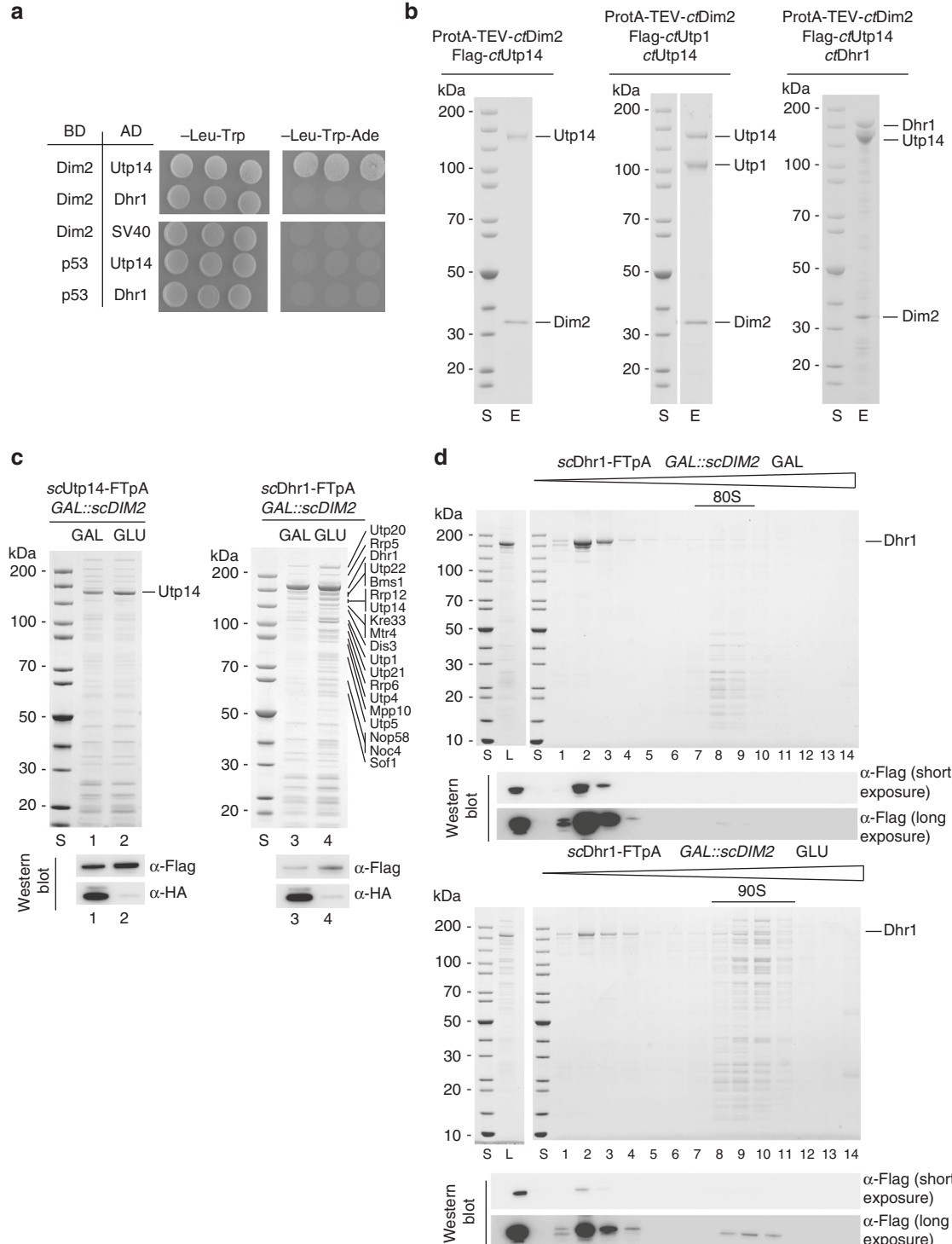

is viable but exhibits a slight growth defect (in particular at higher temperatures), pointing to a mildly impaired Dim2 function in vivo (Supplementary Fig. 6). When Dim2 (W113D) was combined with a viable Utp14 mutant (Utp14$_{multi-sup}$), which mapped in a highly conserved Utp14 C-motif important for Dhr1-Utp14 interaction[19], the double mutants were synthetically lethal (Supplementary Fig. 6). Thus, genetic analysis revealed that Dim2 is functionally linked to Utp14, an activator of Dhr1.

To further investigate the link between Dim2 and the Utp14-Dhr1 array, we performed affinity-purification of Dhr1–FTpA and Utp14–FTpA from yeast cells upon Dim2 depletion. Because the amount and composition of Utp14-associated proteins was not significantly altered if *DIM2* was repressed (Fig. 3c), we concluded that Dim2 is not required for Utp14 recruitment to or association with 90S particles. This is in line with our finding that Utp14 levels identified in semi-quantitative MS were not altered upon Dim2 depletion (see above). In contrast, affinity-purified Dhr1–FTpA from wild-type cells was rather devoid of 90S factors, but this drastically changed when the helicase was isolated from Dim2-depleted cells, causing a massive co-enrichment of the numerous 90S factors (Fig. 3c, Supplementary Fig. 7). Factors of the exosome (e.g., Rrp6, Dis3), an RNA degrading machine, were also found enriched in our purifications under mutant conditions, which is indication that the nuclear exosome is recruited to these mutant 90S particles for turnover of e.g., 5′ ETS rRNA, as previously shown[33].

To directly show that Dhr1 becomes trapped on 90S pre-ribosomes upon Dim2 depletion, we performed sucrose gradient centrifugation. Flag-tagged Dhr1 was purified under conditions of Dim2 expression (galactose) and Dim2 repression (glucose). In the presence of Dim2, the major pool of Dhr1 was found as free protein on top of the sucrose gradient, while only a small amount of Dhr1, barely detectable by western blotting, was associated with fractions, which appear to contain contaminating 80S ribosomes. Under conditions of Dim2 repression (glucose), however, the free pool of Dhr1 on top of the gradient decreased, and instead more Dhr1-Flag was associated with 90S particles and its typical 90S factors (Fig. 3d). Together, these data support a role of Dim2 in activating Dhr1 via the bridging factor Utp14, which in consequence could trigger progression in the 90S-to-pre-40S transition (Discussion).

**Krr1 associates with Rps14 and Rps1 on the 90S pre-ribosome**. As mentioned above, we hypothesized that the second KH-like density on the thermophile 90S pre-ribosome, with contacts to the ribosomal proteins Rps14 and Rps1, is *ct*Krr1. Therefore, we sought to demonstrate these interactions through biochemical reconstitution experiments. For this purpose, ProtA-tagged *ct*Krr1, Flag-tagged *ct*Rps14 and untagged *ct*Rps1 were co-expressed in yeast, followed by split-tag tandem affinity-purification. Strikingly, this allowed the isolation of a stoichiometric *ct*Krr1–Rps14–Rps1 complex (Fig. 4a, upper panel), reinforcing our finding that *ct*Krr1 is docked to the 90S pre-ribosome by binding to Rsp14–Rps1 at the undeveloped platform region.

Because Krr1 occupies a central position at the nascent platform, we investigated how Krr1 contributes to the formation of the platform. We have recently observed a two-hybrid interaction between *ct*Krr1 and *ct*Fap7[37], an assembly chaperone with dual ATPase/adenylate kinase activity, which facilitates incorporation of Rps14 into the pre-ribosome[38–40]. Based on this information, we were able to reconstitute an interaction between *ct*Krr1 and *ct*Fap7, by co-expression of these thermophilic orthologs in yeast and subsequent affinity-purification. We found that endogenous yeast Rps14 was also incorporated into a hybrid complex, *ct*Krr1–*ct*Fap7–*sc*Rps14 (Fig. 4a). As the C-terminus of Rps14 in a complex with Fap7 excludes binding to rRNA (Fig. 4a, lower panel), the Krr1–Rps14–Fap7 complex might be an assembly intermediate, from which Fap7 dissociates upon incorporation of Krr1–Rps14 into the 90S pre-ribosome (see Discussion).

Rps26 (eS26) is another ribosomal protein, which in the mature 40S subunit is directly bound to Rps14[41]. Recently, it was reported that Rps26 assembles into the 90S pre-ribosome at an early stage[42,43]. However, if Krr1 is bound to Rps14 in the 90S particle, Rps26 recruitment appears unlikely due to steric hindrance (Fig. 4b, lower panel). This is consistent with our finding that Rps26 was neither detected in the 90S pre-ribosome by biochemical means, nor in the cryo-EM map[7]. Therefore, we re-investigated the association of Rps26 with different pre-ribosomal particles. This revealed that Rps26 is not present in 90S/pre-40S (Enp1 as bait) or pre-40S preparations (Rio2 as bait), but can be readily detected in mature 40S subunits (Asc1 as bait; Fig. 4b, upper panel). Hence, these data reinforce the idea that Rps26 assembles into the nascent 40S subunit most likely after Dim2 is released from the pre-40S particles during a late maturation step.

**Krr1 is required for Rps14, Rps1, and UTP-C recruitment**. Finally, we sought to analyze the consequences of Krr1 depletion on the assembly of the 90S pre-ribosome. For this reason, we constructed an auxin-inducible Krr1 degron strain[44], which allowed fast and efficient Krr1 degradation during a 90-min incubation in an auxin-containing medium (Supplementary Fig. 8a). Subsequently, the UTP-A factor Utp10–FTpA and the UTP-B factor Utp1-FTpA were affinity-purified and analyzed by SDS-PAGE and Coomassie staining (Fig. 5a). We observed that

**Fig. 3** Dim2 interacts with Utp14, an activator of Dhr1 helicase. **a** 2-hybrid interaction between Dim2 and Utp14. Yeast 2-hybrid analysis of three individual transformants, harboring the indicated bait (BD) and prey (AD) plasmids. Transformants were analyzed by growth on SDC-Leu-Trp (plating efficiency) and SDC-Leu-Trp-Ade plates (strong interaction). SV40 and p53 served as negative controls. The growth was analyzed after 2 days at 30 °C. **b** Reconstitution of *ct*Dim2-Utp1-Utp14-Dhr1 complexes. The indicated dimeric *ct*Dim2-Utp14 and trimeric *ct*Dim2-Utp1-Utp14 and *ct*Dim2-Utp14-Dhr1 complexes using *Chaetomium thermophilum* proteins were assembled by co-expression in yeast followed by split-tag tandem affinity-purification. The reconstituted complexes were analyzed via SDS-PAGE (4–12%) and Coomassie staining. **c** Yeast strains *GAL::scDIM2* with integrated *sc*Dhr1-FTpA or *sc*Utp14-FTpA were grown in galactose (GAL) medium or shifted for 8 h to glucose (GLU) containing medium before the indicated bait proteins were tandem affinity-purified in two consecutive steps involving the ProtA—in the first and the Flag-tag in the second step. TCA-precipitated Flag eluates were analyzed by SDS-PAGE (4–12%) and Coomassie staining (upper panel) or western blotting (lower panel) using the indicated antibodies. The indicated protein bands were identified via mass spectrometry. **d** Final eluates derived from *sc*Dhr1-FTpA tandem affinity-purification under conditions of Dim2 expression (GAL) and depletion (GLU, 8 h), respectively, were analyzed by sucrose gradient (15–40%) centrifugation. Gradient fractions (1–14) were TCA-precipitated and analyzed by SDS-PAGE (4–12%) and Coomassie staining (upper panels) and western blotting against Dhr1-Flag (lower panels, long and short exposure times are depicted). The experiments were performed at least twice with consistent results. Uncropped images are shown in Supplementary Fig. 9. L load, S molecular weight protein standard, E eluate

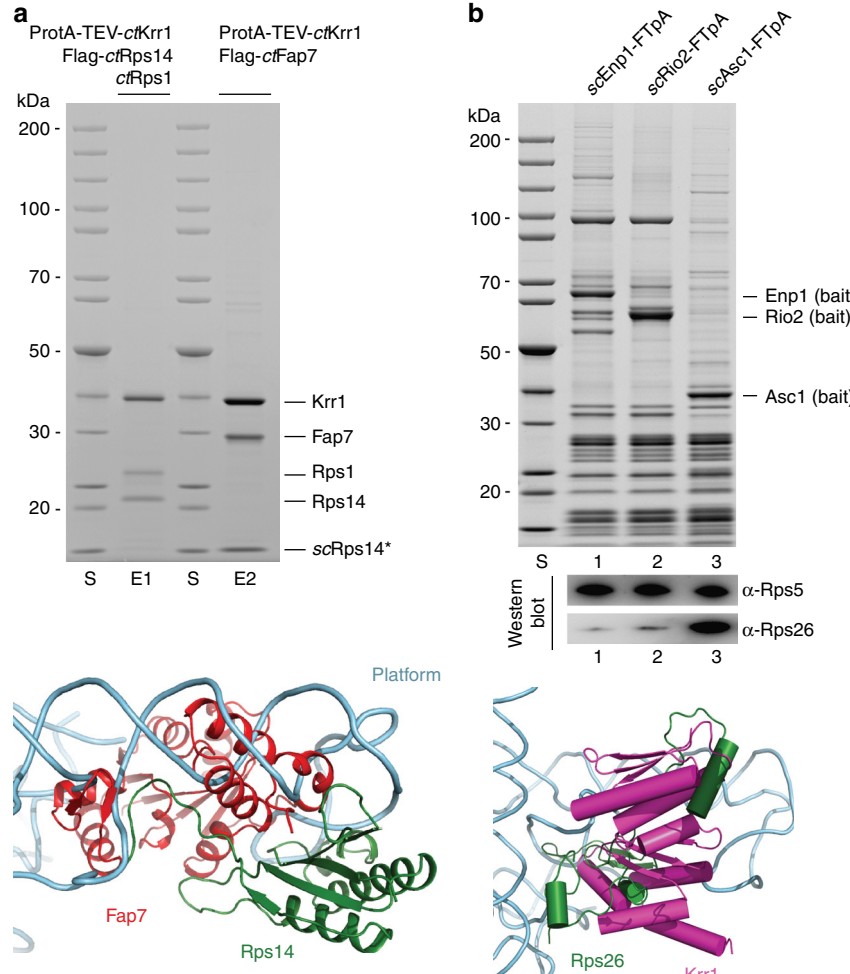

**Fig. 4** Interaction network of Krr1 on the 90S pre-ribosome. **a** *ct*Krr1 forms a complex with ribosomal proteins Rps14 and Rps1. The indicated heterotrimeric *ct*Krr1-*ct*Rps14-*ct*Rps1 and hybrid complex *ct*Krr1-*ct*Fap7-*sc*Rps14, which includes the endogenous yeast ribosomal protein Rps14, were assembled by co-expression of the indicated *Chaetomium thermophilum* factors in yeast followed by split-tag tandem affinity-purification. The reconstituted complexes were analyzed via SDS-PAGE (4–12%) and Coomassie staining. The labeled protein bands were identified by mass spectrometry. The depicted crystal structure of Fap7 (red) bound to Rps14 (green) (lower panel) shows how Fap7 binding clashes with binding of Rps14 to rRNA helix 23 (light blue) within the 90S pre-ribosome. Superimposition was carried out based on the Rps14 proteins from both the isolated platform of the 90S pre-ribosome (PDB: 5JPQ) and the crystal structure of the Rps14-Fap7 complex (PDB: 4CW7). For better visualization, only the crystal structure of Rps14-Fap7 and the rRNA of the platform were shown in cartoon representation. **b** Binding of ribosomal protein Rps26 clashes with the Krr1 binding site on the 90S particle. Tandem affinity-purification of FTpA-tagged Enp1, Rio2 and Asc1. Eluates were TCA-precipitated and analyzed by SDS-PAGE (4–12%) and Coomassie staining (upper panel) or western blotting (lower panel) using indicated antibodies. The lower panel shows a superimposition of Rps14 proteins from both the isolated platform of the 90S pre-ribosome (PDB:5JPQ) and the crystal structure of the mature 40S (PDB: 4V5O). For a better visualization, only Rps26 (green), Krr1 (pink) and the rRNA (blue) of the 40S platform were depicted in the cartoon representation. The experiments were performed at least twice with consistent results. Uncropped images are shown in Supplementary Fig. 9. S molecular weight protein standard, E1 and E2 eluates 1 and 2

one band in the low molecular weight range of the gel, identified as Rps1, was significantly reduced in 90S particles purified from the Krr1-depleted strains, whereas the slightly faster migrating Rps5 band was less affected (Fig. 5a, upper panel). Western blot analysis revealed that also Rps14 was reduced upon Krr1 depletion, whereas Rps5 and Rps8 were less affected (Fig. 5a, lower panel). These findings point to a role of Krr1 recruiting Rps1-Rps14 to the 90S pre-ribosome.

In addition, we noticed that another protein band in the high molecular weight range of the gel, which was identified as Utp22, was specifically absent from 90S particles under conditions of Krr1 depletion (Fig. 5a, see also magnified area of this part of the gel). Western blot analysis and semi-quantitative MS confirmed and extended this observation that not only Utp22, but also its interaction partner Rrp7, both together constituting the UTP-C

module[12,45], were reduced upon Krr1 depletion (Supplementary Fig. 8b). Consistent with this in vivo depletion data, we were able to reconstitute a stable *ct*Utp22–Rrp7–Krr1 heterotrimeric complex through affinity-purification of the thermophilic proteins co-expressed in yeast (Fig. 5b).

Notably, also the low abundant Rok1 helicase was strongly co-depleted upon Krr1 degradation (Supplementary Fig. 8b), which could point to a direct interaction of UTP-C with this ATP-dependent remodeling enzyme[46]. However, Rrp5, previously shown to recruit the UTP-C complex[12], was only marginally affected (Fig. 5a, Supplementary Data 1).

Altogether, the data indicate that Krr1 performs a dual role in early ribosome biogenesis, acting as a placeholder for Dim2, and recruiting Rps1, Rps14, and the UTP-C module to the platform region of the 90S pre-ribosome.

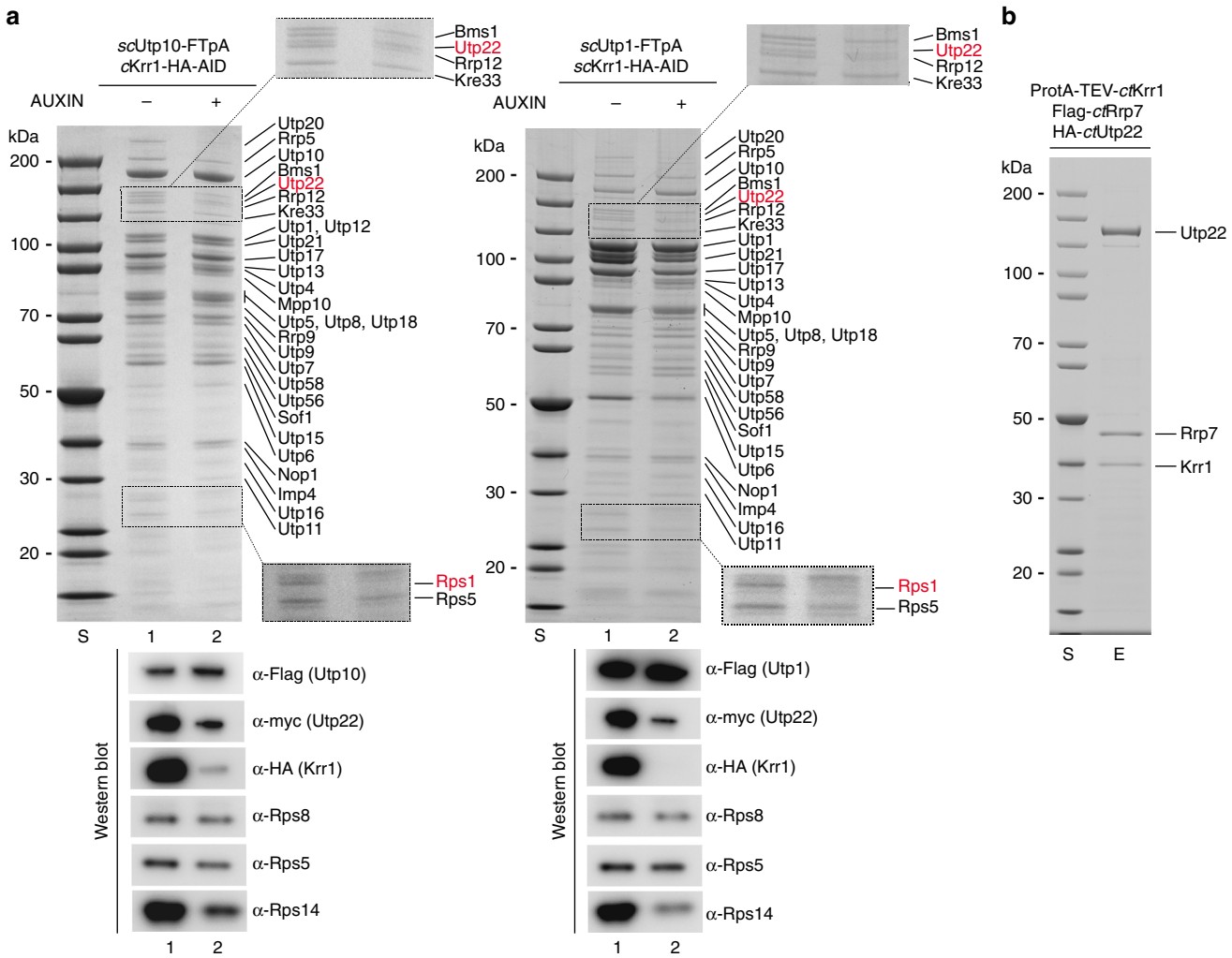

**Fig. 5** Krr1 is required for recruitment of Rps1, Rps14 and UTP-C to the 90S pre-ribosome. **a** Depletion of Krr1 causes recruitment defects of the UTP-C complex, Rps14, and Rps1 to the 90S pre-ribosome. A yeast Krr1 degron strain (scKrr1-HA$_3$-AID) expressing integrated scUtp10-FTpA and scUtp22-myc were grown in YPD medium, before degradation of Krr1 was induced by addition of indole-3-acetic acid (Auxin) to the medium and further growth for 90 min. Subsequently, scUtp10-FTpA was affinity-purified and the eluates derived from the non-depleted and Krr1-depleted strains, respectively, were analyzed by SDS-PAGE (4–12%) and Coomassie staining (upper panel), or western blotting (lower panel) using the indicated antibodies. Dashed box is a zoom of the respective gel region to better display the Utp22/Rps1 depletion. Indicated proteins were identified via mass spectrometry. **b** Krr1 forms a complex with Utp22 and Rrp7 in vitro. The indicated ctUtp22-Rrp7-Krr1 complex was assembled by co-expression of the corresponding *Chaetomium thermophilum* factors in yeast followed by split-tag tandem affinity-purification. The final eluate was analyzed by SDS-PAGE (4–12%) and Coomassie staining. Labeled bands were verified by mass spectrometry identification. The experiments were performed at least twice with consistent results. Uncropped images are shown in Supplementary Fig. 9. S molecular weight protein standard, E eluate

## Discussion

The recent 3D structures of the 90S pre-ribosome from the thermophilic fungus *C. thermophilum* and yeast *Saccharomyces cerevisiae* have provided deep insight into how a myriad assembly factors form a mold-like scaffold on this early assembly intermediate[7–9]. In this study, we focused on two structurally related KH domain proteins, Dim2 and Krr1, present on the 90S pre-ribosome. Based on in vitro reconstitution and in vivo genetic data, we can suggest how Krr1 and Dim2 could participate stepwise in the early ribosome maturation pathway.

One of our key findings concerns the role of Dim2 in 90S biogenesis, which is based on its physical interaction with the UTP-B module. Through this association, Dim2 is strategically positioned on the 90S pre-ribosome to receive and transmit information about the correct assembly status of the nascent ribosome (Fig. 6). UTP-B is a module that exhibits "antenna"-like properties to collect information about the assembly status from different areas of the 90S particle. One of these relay-like interactions concerns the U3 snoRNP, which reaches deeply into the 90S core by base-pairing to several complementary sites within both 5′-ETS and 18S rRNA[47]. The formation and stabilization of these RNA heteroduplexes is multifaceted and dynamic, requiring additional 90S factors and modules such as Mpp10–Imp3–Imp4[48,49]. Following correct 90S assembly and coordinated pre-rRNA processing at different sites, this information can be transmitted from UTP-B via Dim2 to nearby Utp14, which functions as a co-factor and activator of Dhr1's helicase activity. In this way, Dhr1 could induce dissociation of the U3 snoRNP from its pre-rRNA binding sites by unwinding RNA heteroduplexes, with all the consequences for the 90S-to-pre-40S transition[50]. This proposed mechanism nicely explains why Dhr1 becomes stuck on the 90S pre-ribosome upon Dim2 depletion and why Utp14 mutations impair binding and activation of Dhr1[19].

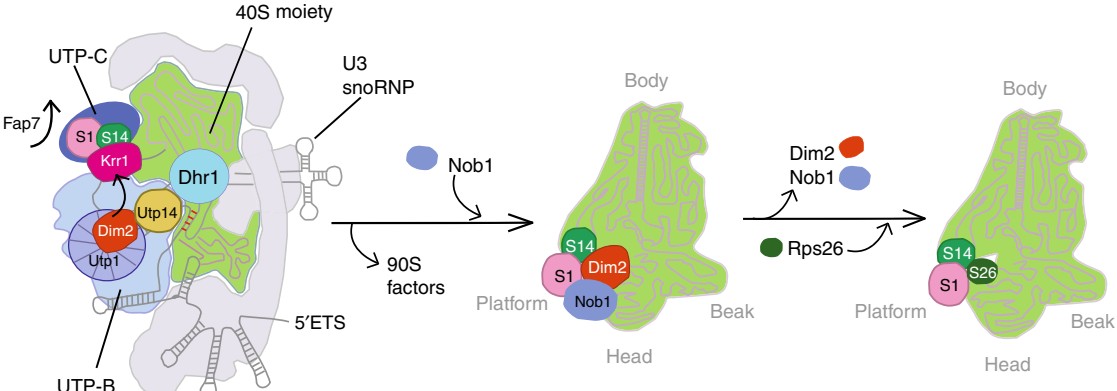

**Fig. 6** Model of how Krr1 and Dim2 function during 90S-to-pre-40S subunit transition. Krr1 is depicted at a distinct site on the 90S pre-ribosome, where it binds adjacent to Rps14 and Rps1 and helps to recruit of UTP-C complex (Utp22 and Rrp7). It is possible that Krr1 facilitates early incorporation of Rps14 and Rps1 into the 90S pre-ribosome with the help of Fap7, which dissociates during this step, thereby making the rRNA binding site on Rps14 available. In contrast, Dim2 is recruited to the 90S pre-ribosome by binding to the UTP-B subunit Utp1, depicted as β-propeller protein. The correct positioning of Dim2 at this site allows the subsequent activation of Dhr1 by its activator Utp14. The stimulation of Dhr1's helicase activity leads to progression in ribosome biogenesis by removal of the U3 snoRNA, a requirement for pseudoknot formation, $A_2$ cleavage, and the dismantling of the 90S pre-ribosome. Subsequently, the pre-40S moiety is set free with Dim2 undergoing relocation to overtake the previously occupied binding site of Krr1 at the platform/head region of the pre-40S ribosome, with contact to Rps14-Rps1. The positioning of Dim2 in this area allows recruitment of the endonuclease Nob1 at a close-by position. After dissociation of Dim2 and D-site cleavage by Nob1, Rps26 can join the 40S subunit

After dismantling the 90S pre-ribosome and liberation of the pre-40S moiety, Dim2 continues its job as a biogenesis factor in the subsequent maturation steps, in which it functions at the platform/neck/head region of the derived pre-40S particle. At present, it is unclear whether Dim2 completely dissociates following 90S disassembly, or whether it relocates instantly from its previous UTP-B-binding site to the nearby site occupied by Krr1 within the 90S pre-ribosome, carrying along other structural elements of the 40S subunit (Fig. 5). However, it is also possible that Dim2 first dissociates from the 90S pre-ribosome upon dismantling, but re-joins later upon pre-40S particle emergence.

Our finding that Krr1 forms a stable complex with Rps14 and Rps1 and Krr1 depletion leads to a reduction of Rps14 and Rps1, but not of other tested ribosomal proteins such as Rps5 and Rps8 on the 90S pre-ribosome not only confirms the positioning of Krr1 within the 90S cryo-EM structure, but it also allows us to speculate that Krr1 mediates the early incorporation of Rps14 and Rps1[51,52] or helps developing this functionally important area further. This notion is also supported by another finding that the Krr1–Rps14 complex can associate with Fap7, a nucleolar biogenesis factor with a dual adenylate kinase/ATPase activity implicated in delivering Rps14 to the pre-ribosome[38–40,42]. The stage at which Fap7 acts has remained controversial since it was suggested that the Fap7–Rps14 interaction is also required for a late step of pre-40S maturation. Our findings could indicate that Fap7 might have a role in promoting the assembly of Rps14 and perhaps Krr1 into the 90S pre-ribosome, such as that Krr1–Rps14–Fap7 could be delivered as an assembly intermediate to the 90S particle. Alternatively, Krr1 might bind first to the 90S pre-ribosome, which would subsequently allow the Fap7–Rps14 complex to be recruited to the 90S. Subsequently, upon Fap7-mediated ATP hydrolysis, the chaperone could dissociate, thereby allowing the C-terminus of Rps14, previously shown to interact with Fap7[38], to become engaged in rRNA contacts. In this context, it is worth mentioning that haplo-insufficiency of Rps14 in human cells has been linked to the genetic disorder 5q-syndrome, a sub-type of the myelodysplastic syndrome characterized by a defect in erythroid differentiation with onset of acute myeloid leukemia[53], and that human Fap7 (hCINAP) is highly expressed

in cancer cells and promotes malignancy by selectively modulating the cancer-associated translatome[54].

Rps26, a ribosomal protein found in mature 40S subunits in close proximity to Rps14, was recently reported to assemble early into the 90S pre-ribosome[42]; however, we find this incompatible with our purification data, from which we see a clear enrichment of Rps26 only on mature 40S subunits. Because the Rps26 moiety clashes with the Krr1 binding site on the 90S particle and with the Dim2 moiety on pre-40S particles, we conclude that Rps26 assembles late into the nascent 40S subunit, most likely following release of Dim2 from the cytoplasmic pre-40S particles. This would finally reveal the binding site for Rps26 next to Rps14 on the 40S subunit. Such a scenario is in agreement with previous findings that Flag-tagged Rps26 mainly co-purifies 18S rRNA, indicating a late association of Rps26 to the SSU[55]. Additionally, this finding is further supported by the fact that Rps26 was not significantly detected in MS studies of 90S or pre-40S particles[7,9,56].

Our data allow us to speculate how another key module of the 90S pre-ribosome, the UTP-C complex members Utp22 and Rrp7, could be recruited to the 90S pre-ribosome via the help of Krr1. Interestingly, the recently published cryo-EM map of the *S. cerevisiae* 90S particle revealed that Utp22 and Rrp7 are located in close proximity to Krr1 and Rps1-Rps14 without a direct contact between Krr1 and Utp22-Rrp7[9]. However, not all parts of Krr1 and the UTP-C module could be modeled into the 90S structure. Based on our findings that Krr1-Utp22-Rrp7 form a robust complex, it is conceivable to assume that Krr1 interacts directly with Utp22, Rrp7 or both within the context of the 90S pre-ribosome. Taken all this data together, the location of Krr1 at the evolving head-platform area of the 90S pre-ribosome explains well how this KH domain protein together with neighboring ribosomal proteins such as Rps14 and Rps1 could play a role for recruitment of the UTP-C complex to this site of the 90S pre-ribosome.

Last but not least, the link of UTP-C to regulatory factors, such as Ifh1, that coordinate ribosome protein gene transcription with pre-rRNA transcription and ribosome assembly[57], makes a possible role for Krr1 in the coordination of ribosome assembly with other cellular pathways very intriguing. Consistent with this,

human Krr1 has been implicated in maintenance of pluripotent stem cells[58]. Further work on the human system might reveal whether conserved interactions between Krr1/UTP-C and the factors that link UTP-C with other cellular processes play a role in those processes.

## Methods

**Plasmid constructs**. All yeast and *Escherichia coli* expression plasmids used in this study were constructed following standard protocols. *E. coli* DH5α was used for cloning and plasmid propagation. All DNA fragments amplified by PCR were sequenced and proven to be correct. Plasmids used in this study are listed in Supplementary Table 1.

**Yeast strain generation**. Methods for gene disruption and genomic tagging of proteins in *S. cerevisiae* were described previously[59,60]. *S. cerevisiae* strains used in this study are derived from W303 and are listed in Supplementary Table 2. Yeast-two-hybrid analysis was carried out with strain PJ69-4A.

**Recombinant protein expression in *E. coli***. Purified proteins used in Nob1-Dim2-binding assays and for size-exclusion chromatography were expressed in *E. coli* BL21 codon plus (DE3) cells (EMD Millipore). Expression was induced when the cell culture reached an optical density 600 ($OD_{600}$) of 0.4 by addition of 0.1 mM IPTG. Cells were collected after 4 h of induction at 23 °C. Cell lysis was carried out with a microfluidizer (Microfluidics) and purified in batches with respective affinity resins (see below).

**Purification and size-exclusion chromatography**. Plasmids petMBP MBP-*ct*DIM2; pet24a $HIS_6$-*ctNOB1 1-354*; and petDuet $HIS_6$-*ctNOB1 250-355*, *ctNOB1 1-169*, pet24d *ctDIM2*; petDuet $HIS_6$-*ctDIM2*, *ctNOB1* and petDuet $HIS_6$-*ctDIM2*, *ctNOB1 W267G* were co-transformed and propagated in *E. coli* BL21. Cells were resuspended in purification buffer NaCl-150 (20 mM Hepes pH 7.5, 150 mM NaCl, 1.5 mM $MgCl_2$, 0.01% Igepal, 1 mM DTT). After cell lysis utilizing a microfluidizer (Microfluidics), samples were incubated with SP Sepharose (Sigma-Aldrich) for 30 min in order to reduce ribosomal contamination. Beads were washed with 20-fold excess of purification buffer NaCl-150, and proteins were eluted with purification buffer NaCl-500 (20 mM Hepes pH 7.5, 500 mM NaCl, 1.5 mM $MgCl_2$, 0.01% Igepal, 1 mM DTT). Salt concentration of the eluate was reduced to 250 mM NaCl by addition of equal volumes of purification buffer NaCl-0 (20 mM Hepes pH 7.5, 0 mM NaCl, 1.5 mM $MgCl_2$, 0.01% Igepal, 1 mM DTT). Subsequently, the eluate was purified via NiNTA (Macherey-Nagel) for 1 h with addition of 10 mM imidazole in order to further minimize background binding. Bound complexes were eluted with purification buffer NaCl-150 containing additional 250 mM imidazole. Size-exclusion chromatography was carried out for the MBP-*ct*Dim2 - $HIS_6$-*ct*Nob1 1-354 complex. Therefore, the Ni-NTA eluate (see above) was loaded on a Superdex-200 10/300 GL column attached to the AKTA basic system (GE Healthcare) and equilibrated with purification buffer NaCl-150. Proteins were separated at a flow rate of 0.6 ml per min at 4 °C. Indicated fractions were analyzed by SDS-PAGE (4–12% gradient polyacrylamide gels; NuPAGE, Invitrogen) and stained with colloidal Coomassie (Roti-Blue, Roth).

**Tandem affinity-purification**. Genomically Flag-TEV-ProtA (FTpA) tagged yeast strains (Supplementary Table 2) were grown in YPD medium to an $OD_{600}$ of 2.5. For Dim2 depletion experiments, Dim2 shuffle strains containing Ycplac111 *pGAL-DIM2* were grown in YPD for 8 h to an $OD_{600}$ of 2.5. For Krr1 depletion experiments, W303α *KRR1-HA2-AID* was grown in YPD to an $OD_{600}$ of 2.5 with addition of 500 µM auxin (IAA) for 90 min prior collecting. Frozen yeast pellets were resuspended in TAP-buffer-"lysis" (100 mM NaCl, 50 mM Tris-HCl pH 7.5, 1.5 mM MgCl2, 0.1% Igepal, 5% glycerol, 1 mM DTT). After cell lysis via bead beating (Pulverisette, Fritsch), the samples were cleared by centrifugation at 35,000×*g* for 20 min at 4 °C. The first affinity-purification step was carried out in batch with immunoglobulin G Sepharose 6 Fast Flow (GE Healthcare) for 2 h at 4 °C. After excessive washing with TAP-buffer (100 mM NaCl, 50 mM Tris-HCl pH 7.5, 1.5 mM MgCl2, 0.01% Igepal, 5% glycerol, 1 mM DTT), the samples were eluted by addition of TEV protease for 1 h at 16 °C. Eluates were further purified utilizing Flag agarose beads (ANTI-FlagM2 affinity gel, Sigma-Aldrich) for 45 min at 4 °C. After washing, the samples were eluted with TAP buffer containing 1× Flag peptide or 3× Flag peptide (Sigma-Aldrich) for single or triple flag-tagged bait protein, respectively. Eluates were TCA-precipitated (final concentration 12.5%) and resuspended in SDS-sample buffer. Proteins were separated utilizing 4–12% gradient polyacrylamide gels (NuPAGE, Invitrogen) and stained with colloidal Coomassie (Roti-Blue, Roth). Western blot analysis was performed according to standard protocols and developed with ImageQuant LAS 4000, GE. Antibodies and dilutions used in this study are displayed below.

**In vitro binding assays**. Prey and bait proteins were expressed as described above in *E. coli* BL21 cells. Cells were resuspended in purification buffer NaCl-150 (20 mM Hepes pH 7.5, 150 mM NaCl, 1.5 mM MgCl2, 0.01% Igepal, 1 mM DTT).

After cell lysis utilizing a microfluidizer (Microfluidics), GST-tagged bait proteins were immobilized on glutathion-agarose beads and NiNTA purified $HIS_6$-tagged prey proteins (see Purification and size-exclusion chromatography of Dim2-Nob1 complexes section) were added in ~10-fold molar excess. Additionally *E. coli* BL21 lysate was added to compete for unspecific binding. Binding reactions were incubated for 45 min at 4 °C. Beads were washed with excess purification buffer NaCl-150 and subsequently eluted in SDS sample buffer by incubation for 4 min at 95 °C. Eluates were analyzed by SDS-PAGE and Coomassie staining.

**Antibodies**. The following antibody dilutions (in PBS + 0.05% Tween + 2% (w/v) milk powder) were used in this study: anti-ProtA (peroxidase anti-peroxidase, Sigma-Aldrich −P1291, 1:3000), anti-Flag (m2-peroxidase conjugate, Sigma-Aldrich A-8592, 1:2000), anti-HA (Covance/HISS diagnostics, MMS-101R, 1:1000), anti-myc (Millipore, cat. no.: 06-549, 1:3000), anti-Rps5 (Santa Cruz Biotechnology, C-200, 1:500) (the antibody was tested in our laboratory for specificity)[61], anti-Rpl5[62] (1:500), anti-Rps8[63] (1:5000), anti-Nob1[27] (1:2000), anti-Rps26/Tsr2[43] (1:2000), anti-Rps14/Fap7[42] (1:2000) and secondary antibodies goat anti-mouse (Bio-Rad -170-6516, 1:3000), donkey anti-goat (Dianova, 705-075-147), goat anti-rabbit (Bio-Rad -170-6515, 1:2000). Validation of commercial primary antibodies is provided on the manufacturers' website.

**Split-tag affinity-purification of protein complexes**. ORFs of the *C. thermophilum* derived proteins (listed below) were cloned into yeast expression plasmids pMT_LEU2_pA-TEV (PGAL1-10, 2µ, LEU2), pMT_TRP1_Flag3 (PGAL1-10, 2µ, TRP1), pMT_URA3 (PGAL1-10, 2µ, URA3), or pMT_URA3_HA3 (PGAL1-10, 2µ, URA3) allowing galactose induced expression in yeast[64] with the following combinations: ProtA-TEV-*ct*Dim2, Flag3-*ct*Utp1; ProtA-TEV-*ct*Dim2, Flag3-*ct*Utp14; ProtA-TEV-*ct*Dim2, Flag3-*ct*Utp1, *ct*Utp14; ProtA-TEV-*ct*Dim2, Flag3-*ct*Utp14, HA3-Dhr1; ProtA-TEV-*ct*Krr1, Flag3-*ct*Rps14, Rps1; ProtA-TEV-*ct*Krr1, Flag3-*ct*Fap7; ProtA-TEV-*ct*Krr1, Flag3-*ct*Rrp7, HA3-Utp22, which were co-transformed in *S. cerevisiae* strain W303 (*MAT alpha, ura3-1, trp1-1, his3-11,15, leu2-3,112, ade2-1, can1-100, GAL+*). Derived yeast transformants with the various plasmid combinations were grown in 1 liter raffinose medium (SRC) lacking the respective amino acids in order to keep selective pressure, at 30 °C to an $OD_{600}$ of 2, before addition of 1 liter 2× galactose medium (2× YPG) to induce expression of the respective *C. thermophilum* proteins. Cells were collected 4 h after induction. Cell lysis and purification was carried out as described above (tandem affinity-purification) for all protein complexes except ProtA-TEV-*ct*Krr1, Flag3-*ct*Rrp7, HA3-*ct*Utp22, where an additional RNase/DNase treatment step was necessary in order to minimize ribosomal contaminations. Therefore, after washing, the Flag-bound protein complex was incubated in TAP-buffer containing additional 2.5 mM MnCl2, 0.5 mM CaCl2, 75 µg/ml RNase1 (ThermoFisher Scientific) and 2.5 units DNase1 per ml (ThermoFisher Scientific) for 20 min at 23 °C. Beads were washed again in TAP-buffer and elution was carried out as described above.

**Sucrose gradient analysis**. Tandem affinity-purified Dhr1-FTpA was loaded on a 15–40% (w/v) linear sucrose gradient containing 100 mM NaCl, 50 mM Tris-HCl pH 7.5, 1.5 mM MgCl2, 0.01% Igepal, 1 mM DTT. Gradients were centrifuged at 129,300×*g* (4 °C) in a SW40 rotor (Beckman Coulter) for 16 h. Respective fractions were collected and TCA-precipitated (final concentration 12.5%), the pellets were washed with aceton and subsequently resuspended in SDS-sample buffer. Proteins were analyzed utilizing 4–12% gradient polyacrylamide gels (NuPAGE, Invitrogen) and stained with colloidal Coomassie (Roti-Blue, Roth).

**Semi-quantitative mass spectrometry and data analysis**. For MS analysis, eluates of TAP-purifications were run on 4–12% polyacrylamide-SDS gel (NuPAGE 4–12%, ThermoFisher Scientific) until they migrated 2–2.5 cm into the gel. After staining the gel with colloidal Coomassie Brilliant Blue G250 (Bio-Rad), the stained area of the lane was excised and the probe was analyzed by FingerPrints proteomics (University of Dundee, UK). Proteins were identified by 1D nLC-ESI-MS-MS and raw MS files were analyzed by MaxQuant software[65]. The iBAQ values of detected proteins were normalized against the respective bait protein. For Fig. 2d and Supplementary Fig. 8b, the iBAQ ratios of depleted versus non-depleted iBAQ values of the most abundant 90S ribosomal biogenesis factors (iBAQ values of at least $1 × 10^6$ in the mock control) were analyzed.

**Yeast-two-hybrid analysis**. Yeast-two-hybrid analysis was performed as described[66]. Y2H-strain PJ69-4a (trp1–901, leu2–3,112, ura3-52, his3-200, gal4D, gal80D, LYS2::GAL1–HIS3, GAL2–ADE2, met2::GAL7-lacZ) was co-transformed with the indicated Y2H constructs pGADT7 and pGBKT7 (Supplementary Table 1). pGADT7-SV40 and pGBKT7-p53 served as negative control plasmids. Three to four individual colonies were spotted on SDC-LEU-TRP agar plates to control plating efficiency. Positive interactions are observed by growth on SDC-TRP-LEU-HIS (weak interaction) and SDC-TRP-LEU-ADE (strong interactions). Plates were incubated for 2 days at 30 °C.

**Multiple sequence alignments**. Protein sequences were aligned with T-Coffee[67] and displayed using Jalview[68].

**Figure preparation**. All images displaying electron densities and molecular models were generated with UCSF Chimera[69].

**Data availability**. The authors declare that the data supporting the findings of this study are available within the paper and its supplementary information files, and available from the corresponding author upon request.

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

## Acknowledgements

We are grateful to Dr. E. Thomson for providing plasmid pet24d GST-TEV-*ct*Dim2, helpful advice on the project and reading the manuscript, and M. Thoms for providing plasmids and yeast strains for constructing auxin-regulatable degron strains. We thank V. Panse, K. Karbstein, A. Johnson, and J.L. Woolford for providing antibodies and yeast mutant alleles. We thank P. Ihrig and J. Reichert (Lechner lab) for performing MS analysis (MS facility BZH, Heidelberg).

## Author contributions

M.S. and E.H. designed the study. All experiments were performed by M.S. Semi-quantitative MS data were analyzed by M.S. and J.B. J.B. cloned *ct*Utp14, *ct*Utp1, and *ct*Dim2 in yeast over-expression plasmids and performed initial test purifications. J.C. and R.B. prepared the molecular models. The manuscript was written by M.S. and E.H. All authors discussed the results and commented on the manuscript.

## Additional information

**Competing interests:** The authors declare no competing financial interests.

