## [Peer Review File · Nature Communications]

Reviewers' comments:

Reviewer #1 (Remarks to the Author):

In this manuscript, Strum et al. address the functions of Krr1 and Dim2, two KH-domain proteins with essential roles in the synthesis of 40S ribosomal subunits. At the beginning of the study, the authors explain that the available cryo-EM maps of the 90S pre-ribosome show the presence of KH-like densities in two positions of the structure. However, due to the low-resolution in those regions, the identity of the proteins (presumably Krr1 and Dim2) had not been assigned. After a series of protein-complex reconstitution experiments and two-hybrid assays, they establish that Krr1 is in close contact with Rps4-Rps1, two ribosomal proteins located on the platform in the mature 40S subunit, and that Dim2 interacts the UTP-B subcomplex. This is an interesting finding because Dim2 (together with Nob1) is positioned on the platform in late pre-40S particles. Therefore, it is possible that Krr1 acts as a placeholder of Dim2 in the platform while Dim2 plays some early-assembly function in concert with UTP-B. The authors proceed to explore these ideas and perform pre-ribosome purifications in Dim2- or Krr1- depleted cells combined with protein complex reconstitution assays. They show that Krr1 interacts directly with UTP-C factors, and that Dim2 interacts with Utp14 and Dhr1. They conclude that Krr1 is a placeholder of Dim2 that recruits the UTP-C complex to the 90S particle. It is also concluded that Dim2 is required for the Dhr1 helicase to remove the U3 snoRNP from the 90S pre-ribosome.

This work is a compilation of fragmentary results that, although potentially interesting, do not fully substantiate the conclusions made by the authors. The issues addressed are rather limited, and the experimental approaches and amount of data are insufficient or need further validation. For example, some of the conclusions rely on characteristics of the pre-ribosomes produced in the absence of Krr1 or Dim2, but the data on those particles are very poor. No information is provided about their structures, and no comprehensive compositional analyses were made. Even the data on the loss of UTP-C in Krr1-depleted particles, or on the entrapment of Dhr1 in Dim2-depleted particles are incomplete (no western blot showing the sizes of Dhr1 pre-ribosomes in Fig. 3d; no co-IP of assembly factors with pre-RNAs in Fig. 4c). Other information that might support the proposed functions of Pno1 in early pre-ribosomes, such as the positioning of Utp14 close to Dim2 in the 90S particle or the existence of genetic interactions between Pno1, Utp14, Dhr1 or Utp1, is also missing. The proposal of Krr1 as a placeholder for Dim2 relies on its location in the cryo-EM structure, but there is no data showing that they actually bind the same site on the rRNA.

In addition to the insufficient amount of novel information, the paper exhibits important faults in terms of the motivation for some of the experiments and data presentation. For example, the rationale and interpretation of the experiments in Fig. 1 are not correct. It has been previously established that Dim2 depletion causes a defect in A2 cleavage that blocks the production of pre-40S particles (Vanrobays et al 2004; Woolls et al. 2011). The loss of the interaction of Nob1 with Enp1 or Rio2 is expected because pre-40S particles are not being made, not because Dim2 is important for Nob1 recruitment. Other problems are seen in the pertinence to address the recruitment of Rps26, which was already found to assemble late in the pathway (see Karbstein review, TICB 2013), or in the digression and vague significance of the Krr1-Rps14-Fap7 subcomplex. In general, the results are shown in a disperse, badly-organized and poorly-argued manner. Most of the conclusions derived from each set of results are overstatements combined with considerations or speculations that should go in the discussion section. Previous findings in the field are not properly referenced, and recent data on the recruitment and position in the 90S particle of the proteins under study (Zhang et al. Genes and Dev. 2016; Sun et al. eLife, 2017) are not discussed. It should be stated somewhere that Dim2 is also known as Pno1.

I consider that the paper has no sufficient quality to justify a revision or resubmission for publication in Nature Communications.

Reviewer #2 (Remarks to the Author):

Reviewer's comments, Sturm et al., NCOMMS-17-07747-T

In the present manuscript, Sturm and colleagues characterize the role of Dim2 and Krr1, two ribosome biogenesis proteins, found within pre-90S and pre-40S particles and known to have KH-motifs. Sturm et al base their hypothesis on recently published 90S pre-ribosome crystal structures and cleverly designed a series of biochemical assays - including yeast-two-hybrids, in vitro assays and affinity purifications - to, first, confirm the location of Dim2 and Krr1 proteins within the 90S particles, and to better understand their role within the structure. Moreover, in the process they also identified the role of Dim2 in Nob1 recruitment, the Dim2 binding platform on 90S pre-ribosomes and interaction with Dhr1, as well as recruitment of the UTP-C complex via Krr1 and Fap7.

This manuscript presents very nice work. It provides solid evidence that Dim2 has a role in inducing the 90S to pre-40S transition through interactions with both the UTP-B complex and Utp14. It also supports previous evidence of a role for Dim2 as placeholder for Nob1 in pre-40S maturation. It further shows that Krr1 associates with Rps1 and Rps14 and serves as a binding platform for the UTP-C complex within the 90S pre-ribosome. Overall this manuscript provides

a number of novel insights into hierarchical ribosome assembly making it certainly suitable for publication in Nature Communications.

To make this manuscript appropriate for final publication in Nature Communications, however, the authors need to address the following minor points:

- Page 5, Figure 1a, Western, lanes 2 and 4: The authors should mention the time of Dim2 depletion (in Glucose); also, the depletion seems somewhat variable here and this should be addressed – i.e. how a remaining amount of Dim2 could affect the MS analysis.
- Page 5, 2nd line from bottom of page: “specific dissociation of Nob1 from late pre-40S particles,” – the word ‘dissociation’ should be changed since it would indicate that Nob1 is associated with pre-90S particles prior to Dim2, and when Dim2 does not associate with particles, then Nob1 dissociates. This is not the case since Dim2 is recruited to the complex prior to Nob1. Please change wording accordingly.
- Page 6: “However, other previous data contradict this view, as a two-hybrid interaction between ctDim2 and ctUtp1 (Pwp2), a subunit of the UTP-B module, was observed³⁸, which is distant but not too far away from the KH-like density associated with Rps14 and Rps1 on the 90S pre-ribosome.” The authors should provide a proper distance estimate based on their previous.
- Page 29, mass spec results: In the presented analysis, pre-40S and pre-60S factors were specifically excluded. The authors should provide the complete list of proteins found in MS runs, with semi-quantitative values, in a supplemental table.
- Page 11, line 1: letter missing, should read ‘pre-ribosome’
- Page 27, line 8: letter missing, should read ‘imidazole’
- Page 28, line 7: comma missing, it should read ‘Millipore, Cat.No’
- Page 28, line 8: should read ‘laboratory for specificity’

Reviewer #3 (Remarks to the Author):

This manuscript provides follow up to the recent and remarkable cryo-EM structure of the 90S pre-ribosome that the Beckmann and Hurt labs recently published. That work provided the first large scale structure of the pre-90S but the resolution was not sufficient to assign many of the polypeptides. This manuscript focuses on two of those unassigned assembly factors, the KH domain proteins Dim2 and Krr1. This manuscript uses two-hybrid and in vitro protein interaction

methods to support the assignment of these two KH domain proteins. The authors also examine the consequence of depleting these factors on the composition of the 90S and conclude that Dim2 interacts with Utp14 and suggest that this activates the RNA helicase Dhr1. While the interaction data are all quite clear, overall conclusions are rather modest. Combining the analysis of Dim2 and Krr1 in a single manuscript does add more information, but neither story is very strong and the focus is lost. The evidence for sequential assembly is modest. Consequently, I cannot recommend publishing the work in its current form in Nature Communications.

Major points:

1. I am a little confused about the inability to assign Dim2 and Krr1 as these proteins were assigned in the PDB file associated with their Cell paper Kornprobst et al. These proteins have also been assigned in more recent work from Ye's group. Consequently, the effort to support the assignment of these proteins has lessened impact.
2. The authors make assumptions about their data that may be informed by their own "in house" knowledge but are not evident to the reader. For example, in Fig 1a there are many more bands in the Enp1 pull down from Dim2-depleted cells but despite the claim that there is "massive co-enrichment of normally underrepresented early 90S factors including UTP-A, UTP-B, UTB-C and U3 snoRNP subunits (Fig. 1a)" no evidence is provided identifying any of these proteins.
3. It is not evident from Fig 3c that there is a massive enrichment. From their iBAQ analysis, what is most striking is the enrichment (50-fold) of the nuclear exosome, yet the authors do not comment on this. In addition, the analysis tells us about fold enrichment but not actual stoichiometries of the various factors in the particles. For example, is the exosome stoichiometric or still a minor component when Dim2 is depleted?
4. In general, total depletion of a factor is much more disruptive than expression of a loss-of-binding mutant; depletion of a factor with multiple interaction partners could lead to loss of multiple factors making it difficult to attribute specificity. The authors should map the Dim2-Nob1 interaction and test mutants that fail to interact for their impact on the progression of particle assembly and Nob1 recruitment and cleavage.
5. Semi quantitative mass spec of particles from Dim2-depleted cells identified proteins in addition to Dhr1 that were similarly enriched (Nop9, Nsr1 and Mrd1) and others that were significantly depleted but these were not evident from Coomassie staining. It is not clear why the authors did not explore these other factors. Also, when Krr1 was depleted Utp22 and Rrp7 were lost, but the reduction in the helicase Rok1 was even more dramatic. Why was this ignored?
6. In Fig 3d it looks like the total amount of Dhr1 is reduced in glucose but the ratio of Dhr1 to pre-ribosome in fractions 8-11 is not changed much between the two conditions. This seems counter to the authors' conclusion.

7. In general, the Coomassie stained gels of pull downs appear to show heavy contamination with low molecular weight proteins. Are these mature ribosomal protein contaminants? No negative controls (untagged) are shown for the IPs.

8. The authors should explain how they conclude that Nob1 dissociates in the absence of Dim2 rather than is not recruited to begin with.

Minor points:

8. In Fig 1, if depletion of Dim2 blocks 40S assembly, how is it that Rio2 binds pre-40S lacking Dim2? Does a population of particles slip through the assembly and export pathways lacking Dim2? And, no evidence is provided that the Coomassie-stained bands in the Rio2 pull down are related to pre-40S.

9. Regarding the Nob1 in vitro cleavage, please clarify if cleavage was seen or not seen.

10. The authors mapped the interaction domains of Nob1 and Dim2 in Fig 1C but did not make use of this information to interpret structure or function.

Reviewers' comments and our reply (in red)

Reviewer #1 (Remarks to the Author):

In this manuscript, Strum et al. address the functions of Krr1 and Dim2, two KH-domain proteins with essential roles in the synthesis of 40S ribosomal subunits. At the beginning of the study, the authors explain that the available cryo-EM maps of the 90S pre-ribosome show the presence of KH-like densities in two positions of the structure. However, due to the low-resolution in those regions, the identity of the proteins (presumably Krr1 and Dim2) had not been assigned. After a series of protein-complex reconstitution experiments and two-hybrid assays, **they establish that Krr1 is in close contact with Rps4-Rps1, two ribosomal proteins located on the platform in the mature 40S subunit, and that Dim2 interacts the UTP-B subcomplex. This is an interesting finding** because Dim2 (together with Nob1) is positioned on the platform in late pre-40S particles. Therefore, it is possible that Krr1 acts as a placeholder of Dim2 in the platform while Dim2 plays some early-assembly function in concert with UTP-B. The authors proceed to explore these ideas and perform pre-ribosome purifications in Dim2- or Krr1- depleted cells combined with protein complex reconstitution assays. They show that Krr1 interacts directly with UTP-C factors, and that Dim2 interacts with Utp14 and Dhr1. They conclude that Krr1 is a placeholder of Dim2 that recruits the UTP-C complex to the 90S particle. It is also concluded that Dim2 is required for the Dhr1 helicase to remove the U3 snoRNP from the 90S pre-ribosome.

This work is a compilation of fragmentary results that, although potentially interesting, do not fully substantiate the conclusions made by the authors. The issues addressed are rather limited, and the experimental approaches and amount of data are insufficient or need further validation.

For example, some of the conclusions rely on characteristics of the pre-ribosomes produced in the absence of Krr1 or Dim2, but the data on those particles are very poor. No information is provided about their structures, and **no comprehensive compositional analyses were made.**

These pre-ribosomal particles have been further analyzed by semi-quantitative mass spectrometry and sucrose gradient centrifugation. As an example, we analyzed the Dim2-depleted pre-ribosomal particles purified via the Enp1-bait by sucrose gradient centrifugation and determined the major, i.e. Coomassie-stainable bands by mass spectrometry. These new data are shown in the revised Supp. Fig. 2b. Furthermore, we provide a bar diagram (Supplementary Fig. 7), in which the factors associated with Dhr1-FTpA purifications (grouped according to their organization in subcomplexes) are plotted based on their absolute iBAQ values (i. e. stoichiometry). We indicate the protein identities on the particle purifications in revised Figs. 1a, 2c and 4c. In addition, all semi-quantitative mass spectrometry data are now provided as Excel files.

Even the data **on the loss of UTP-C in Krr1-depleted particles**, or on the entrapment of **Dhr1** in Dim2-depleted **particles are incomplete** (no western blot showing the sizes of Dhr1 pre-ribosomes in Fig. 3d; no co-IP of assembly factors with pre-RNAs in Fig. 4c).

To better indicate the loss of UTP-C in Krr1-depleted particles, we show a zoom of the relevant area in the SDS-PAGE gel (Fig. 4c), clearly revealing that the Coomassie stainable Utp22 band is largely absent from the Utp10-FTpA preparation upon Krr1 depletion. This data was further verified by Western analysis (anti-myc-Utp22) and by semi-quantitative mass spectrometry of the

entire preparation.

Regarding Dhr1 in Dim2-depleted particles, we show Western blot detection of the Dhr1 bait on the sucrose gradient, which clearly indicates trapping of Dhr1 in 90S pre-ribosomal particles (revised Fig. 3d).

Other information that might support the proposed functions of Pno1 in early pre-ribosomes, such as the positioning of Utp14 close to Dim2 in the 90S particle or the **existence of genetic interactions** between **Pno1, Utp14, Dhr1 or Utp1**, is also missing.

We now have performed such genetic analyses, revealing that *DIM2* and *UTP14* are genetically linked. For this experiment, we used a point mutation in the KH1-like domain of Dim2 (W113>D; generated in the course of these studies), which exhibited a mild growth defect. When Dim2 W113>D was combined in with a mutation in Utp14, generated in the Johnson lab (called Utp14_{mult-sup} and mapping in a highly conserved Utp14 C-motif important for Dhr1-Utp14 interaction; see *Mol. Cell. Biol.* 2016, 36, 965-978), a synthetic lethal phenotype was generated (revised Supp. Fig. 6). This new data shows that our *in vitro* data obtained with *Chaetomium thermophilum* factors are functionally relevant in the yeast system, suggestive of conserved interactions.

The proposal of Krr1 as a placeholder for Dim2 relies on its location in the cryo-EM structure, but there is no data showing that they actually bind the same site on the rRNA.

We agree that since there is no high-resolution structure of Krr1 and Dim2 in the 90S and pre-40S particles, respectively, one cannot say with certainty that the binding of the two proteins to the platform site is identical. However, due to the current models derived from several low-resolution cryo-EM structures (Larburu *et al.* *NAR* **44**, 8465-8478 (2016); Johnson *et al.* *Structure* **25**, 329-340 (2017); Strunk *et al.* *Science* **333**, 1449-1453 (2011), it appears fair to say that Krr1 and Dim2 bind in a similar way.

In addition to the insufficient amount of novel information, the paper exhibits **important faults in terms of the motivation for some of the experiments and data presentation**. For example, the rationale and interpretation of the experiments in Fig. 1 are not correct.

We believe that we have carefully interpreted the data in Fig.1. We never said that Dim2 directly recruits Nob1, but we cited previous work from other labs, which concluded that Dim2 is involved in Nob1 recruitment (“This data supports earlier findings that Dim2 plays a role in recruiting Nob1 to the pre-40S particles²⁸”).

However, during revision we have performed additional experiments revealing how Nob1 and Dim2 physically interact with each other (requested by reviewer #3). This new data show that Nob1 uses both its PIN and middle domain (MID) for binding to Dim2. *In vitro* binding assays indicated that deletion of the PIN domain weakens but does not abolish binding to Dim2, whereas removal of the MID domain apparently blocks it. This was shown for both yeast

and *Chaetomium thermophilum* orthologues. In order to assess the *in vivo* role of this interaction, we have deleted the MID-domain from yeast Nob1 (we did not mutate the PIN domain, since it could cause folding problems, thereby acting like a null), which however did neither exhibit a growth defect nor abolish Nob1 Δ MID interaction with late pre-40S particles. Thus, the interaction of Nob1 and Dim2 *in vivo* is complex and appears to depend on redundant mechanisms.

It has been previously established that Dim2 depletion causes a defect in A2 cleavage that blocks the production of pre-40S particles (Vanrobays et al 2004; Woolls et al. 2011).

We have already mentioned and cited one of these papers (Vanrobays; Ref. 31: "To date, Dim2 has been predominantly studied in the context of the late pre-40S maturation pathway though its association with Nob1^{25,28}, although a role in early ribosome assembly has been reported³¹), but now also mention the other paper (Woolls et al.) and in addition state that A2 cleavage was affected.

The loss of the interaction of Nob1 with Enp1 or Rio2 is expected because pre-40S particles are not being made, not because Dim2 is important for Nob1 recruitment.

It is not clear to us why loss of interaction of Nob1 with Enp1 or Rio2 is expected for this reviewer? Actually, very little is known about the exact mechanism how and when Nob1 is recruited to pre-40S particles. Thus, it is still possible that Dim2 recruits Nob1 to pre-40S particles, or at least has a certain role during this step. Our analysis has addressed this by combining *in vitro* binding assays with *in vivo* analyses in yeast.

Actually, the data that Rio2 particle specifically lacks Nob1 is still consistent with a direct role of Dim2 in keeping Nob1 stably associated with pre-40S particles.

This is possible, but our data also suggest that there could be redundant mechanisms for independent targeting of these proteins to the pre-ribosomal particles, which is mentioned in the revised manuscript.

Other problems are seen in the pertinence to address the **recruitment of Rps26, which was already found to assemble late in the pathway (see Karbstein review, TICB 2013)**, or in the digression and vague significance of the Krr1-Rps14-Fap7 subcomplex.

Concerning Rps26 recruitment to early 90S particles, this is not yet firmly established according to the published literature, with so far only circumstantial arguments (e.g. negative data from mass spectrometry etc.). Indeed, the Panse lab continues to argue on the early recruitment of Rps26 with recent publications. Thus, we find it important to show by direct experimental means (cryo-EM and biochemical data) that recruitment of Rps26 to the 90S particles is not possible due to structural reasons. Moreover, we also show by Western blotting using an anti-Rsp26 antibody used in the Panse lab that Rps26 is not present on 90S particles.

Regarding Rps14-Fap7, a late role in pre-40S biogenesis during remodeling of the platform domain and D-site cleavage was previously discussed

(Strunk et al. 2012, Loc'h et al. 2014). Thus, we find this new link of Fap7-Rps14 to Krr1 interesting, suggesting a role of Fap7 in the incorporation of Rps14 into the 90S particles. We believe that showing this data in our manuscript could foster further studies regarding this still poorly explored biogenesis pathway.

In general, the results are shown in a disperse, badly-organized and poorly-argued manner. Most of **the conclusions derived from each set of results are overstatements combined** with considerations or speculations that should go in the discussion section.

We respectfully disagree with these negative statements, as we think that we have not overstated our results, which in all cases are based on solid and highly reproducible biochemical data.

Previous findings in the field are not properly referenced, and recent data on the recruitment and position in the 90S particle of the proteins under study (Zhang et al. *Genes and Dev.* 2016; Sun et al. *eLife*, 2017) are not discussed.

We have discussed and referenced these previous findings.

It should be stated somewhere that Dim2 is also known as Pno1.

This was already mentioned in Supp. Table 3, but is now stated in the abstract as well.

I consider that the paper has no sufficient quality to justify a revision or resubmission for publication in Nature Communications.

In contrast to this reviewer, we feel that many of our findings are original and of high quality, and always reproduced at least twice. This opinion is also reflected by reviewer #2 and #3, who are more positive regarding publication of our data in Nature Communications.

Reviewer #2 (Remarks to the Author):

Reviewer's comments, Sturm et al., NCOMMS-17-07747-T

In the present manuscript, Sturm and colleagues characterize the role of Dim2 and Krr1, two ribosome biogenesis proteins, found within pre-90S and pre-40S particles and known to have KH-motifs. Sturm et al base their hypothesis on recently published 90S pre-ribosome crystal structures and cleverly designed a series of biochemical assays - including yeast-two-hybrids, in vitro assays and affinity purifications - to, first, confirm the location of Dim2 and Krr1 proteins within the 90S particles, and to better understand their role within the structure. Moreover, in the process they also identified the role of Dim2 in Nob1 recruitment, the Dim2 binding platform on 90S pre-ribosomes and interaction with Dhr1, as well as recruitment of the UTP-C complex via Krr1 and Fap7.

This manuscript presents very nice work. It provides solid evidence that Dim2 has a role in inducing the 90S to pre-40S transition through interactions with both the UTP-B complex and Utp14. It also supports previous evidence of a role for Dim2 as placeholder for Nob1 in pre-40S maturation. It further shows that Krr1 associates with Rps1 and Rps14 and serves as a binding platform for the UTP-C complex within the 90S pre-ribosome. Overall this manuscript provides a

number of novel insights into hierarchical ribosome assembly making it certainly suitable for publication in Nature Communications.

To make this manuscript appropriate for final publication in Nature Communications, however, the authors need to address the **following minor points**:

- Page 5, Figure 1a, Western, lanes 2 and 4: The authors should mention the time of Dim2 depletion (in Glucose); also, the depletion seems somewhat variable here and this should be addressed – i.e. how a remaining amount of Dim2 could affect the MS analysis.

The time of depletion (8 hr) is now mentioned in the text. Concerning the variability of Dim2 depletion: in every experiment, the time of depletion was strictly met. However, we observed that a small amount of Dim2 remained after this time (which is typical for GAL-driven gene expression in yeast). We performed semi-quantitative mass spectrometry analyses of the different affinity-purifications at least twice, with always similar iBAQ values for Dim2 and the other co-purified factors.

- Page 5, 2nd line from bottom of page: “specific dissociation of Nob1 from late pre-40S particles,” – the word ‘**dissociation**’ should be changed since it would indicate that Nob1 is associated with pre-90S particles prior to Dim2, and when Dim2 does not associate with particles, then Nob1 dissociates. This is not the case since Dim2 is recruited to the complex prior to Nob1. Please change wording accordingly.

We have changed this term.

- Page 6: “However, other previous data contradict this view, as a two-hybrid interaction between ctDim2 and ctUtp1 (Pwp2), a subunit of the UTP-B module, was observed³⁸, which is distant but not too far away from the KH-like density associated with Rps14 and Rps1 on the 90S pre-ribosome.” The authors should provide a proper distance estimate based on their previous.

We now provide such a value, which is approx. 65Å distant from the KH-like density associated with Rps14 and Rps1 on the 90S pre-ribosome.”

- Page 29, mass spec results: In the presented analysis, pre-40S and pre-60S factors were specifically excluded. The authors should provide the complete list of proteins found in MS runs, with semi-quantitative values, in a supplemental table.

Excel files for these mass spectrometry analysis are now provided.

- Page 11, line 1: letter missing, should read ‘pre-ribosome’
corrected
- Page 27, line 8: letter missing, should read ‘imidazole’
corrected
- Page 28, line 7: comma missing, it should read ‘Millipore, Cat.No’
corrected
- Page 28, line 8: should read ‘laboratory for specificity’
corrected

Reviewer #3 (Remarks to the Author):

This manuscript provides follow up to the recent and remarkable cryo-EM structure of the 90S pre-ribosome that the Beckmann and Hurt labs recently published. That work provided the first large scale structure of the pre-90S but the resolution was not sufficient to assign many of the polypeptides. This manuscript focuses on two of those unassigned assembly factors, the KH domain proteins Dim2 and Krr1. This manuscript uses two-hybrid and in vitro protein interaction methods to support the assignment of these two KH domain proteins. The authors also examine the consequence of depleting these factors on the composition of the 90S and conclude that Dim2 interacts with Utp14 and suggest that this activates the RNA helicase Dhr1. While the interaction data are all quite clear, overall conclusions are rather modest. Combining the analysis of Dim2 and Krr1 in a single manuscript does add more information, but neither story is very strong and the focus is lost. **The evidence for sequential assembly is modest.** Consequently, I cannot recommend publishing the work in its current form in Nature Communications.

Actually, we did not mean 'sequential assembly', rather than Krr1 and Dim2 acting sequentially on the platformsite. In the revised manuscript, we hope that we have better addressed this. Hence, we revised the title to 'Assembly of the 40S subunit platform driven by the subsequent action of KH domain proteins Krr1 and Dim2'.

Major points:

1. I am a little confused about the inability to assign Dim2 and Krr1 as these proteins were assigned in the PDB file associated with their Cell paper Kornprobst et al. These proteins have also been assigned in more recent work from Ye's group. Consequently, the effort to support the assignment of these proteins has lessened impact.

In the Kornprobst et al paper, these 2 densities were only assigned as KH domains. Ye's group indeed assigned it better, but they explicitly stated that the assignment have been dealt cautiously: "some assignments, especially those without high-resolution crystal structures should be considered tentative at the current resolution of cryo-EM map." We have now clearly mentioned these findings from Ye and colleagues in our revised paper. However, it is important in this field and in general to perform additional biochemical and *in vivo* studies to prove these structural models, which was a major aim of our research.

2. The authors make assumptions about their data that may be informed by their own "in house" knowledge but are not evident to the reader. For example, in Fig 1a there are many more bands in the Enp1 pull down from Dim2-depleted cells but despite the claim that there is "massive co-enrichment of normally underrepresented early 90S factors including UTP-A, UTP-B, UTB-C and U3 snoRNP subunits (Fig. 1a)" **no evidence is provided identifying any of these proteins.**

We have included this data based on mass spectrometry of the labeled bands. See revised Figs. 1a, 2c, 3c, 4c and Supplementary Fig. 2b.

3. It is not evident from Fig 3c that there is a massive enrichment. From their iBAQ analysis, what is most striking is the enrichment (50-fold) of the **nuclear exosome, yet the authors do not**

comment on this. In addition, the analysis tells us about fold enrichment but not actual stoichiometries of the various factors in the particles. For example, is the exosome stoichiometric or still a minor component when Dim2 is depleted?

We apologize for not having this discussed in the first version. Indeed, exosome factors were clearly found in our purifications under mutant conditions, which is indication that the nuclear exosome is recruited for turnover of e.g. 5' ETS rRNA, as previously shown (Thoms et al., 2015). To better display the enrichment and stoichiometry of the 90S biogenesis and exosome factors found in the purifications, we included a bar diagram (Supplementary Fig. 7), in which the factors, grouped according to their organization in subcomplexes, were plotted based on their absolute iBAQ values (i.e. stoichiometry). This scheme shows that exosome factors are enriched but less prominently than 90S factors. Moreover, we also labeled the major Coomassie stainable bands of the Dhr1 affinity-purification (Fig. 3c). Finally, we have included all the mass spectrometry data from the different purifications in form of excel files, as requested by reviewer #2.

Regarding the revised manuscript, we have now mentioned in the text that exosome factors are present in these mutant particles.

4. In general, total depletion of a factor is much more disruptive than expression of a loss-of-binding mutant; depletion of a factor with multiple interaction partners could lead to loss of multiple factors making it difficult to attribute specificity. **The authors should map the Dim2-Nob1 interaction and test mutants that fail to interact** for their impact **on the progression of particle assembly and Nob1 recruitment** and cleavage.

As requested, we have performed additional experiments revealing how Nob1 and Dim2 physically interact with each other. This data shows that Nob1 uses predominantly its middle domain (MID), and to a lesser extent the PIN domain, for binding to Dim2. *In vitro* binding assays revealed that deletion of the PIN domain weakens but does not abolish binding to Dim2, whereas removal of the only MID domain apparently blocks the Nob1-Dim2 interaction *in vitro*. This was shown for both yeast and *Chaetomium thermophilum* orthologues (revised Fig. 1c; Supplementary Fig. 3b).

In order to assess the *in vivo* role of this interaction, we have deleted the corresponding MID-domain from yeast Nob1, but did not mutate the PIN domain, since it could cause unpredictable folding problems. Unexpectedly, yeast cells expressing Nob1 Δ MID did not exhibit a growth defect (Supp. Fig. 3c), nor was the association with late pre-40S particles purified via Ltv1 affected (Supp. Fig. 3d). Thus, the robust *in vitro* interaction between Nob1 and Dim2, mediated by Nob1-MID, is less important *in vivo*, suggesting that other (redundant) targeting mechanisms exist that recruit these two assembly factors to the same pre-40S particles.

5. Semi quantitative mass spec of particles from Dim2-depleted cells identified proteins in addition to Dhr1 that were similarly enriched (Nop9, Nsr1 and Mrd1) and others that were significantly depleted but these were not evident from Coomassie staining. It is not clear why the

authors did not explore these other factors.

As correctly pointed out by this reviewer, also other factors were affected, which we did not further follow in this work. In the scope of this study, we mainly concentrated on Dhr1, since our lab has recently found a 2-hybrid interaction between Utp14 (the activator of Dhr1) and Dim2 (Baßler et al. 2016), while we could not detect a direct link between Dim2 and the other factors mentioned above.

Also, when Krr1 was depleted Utp22 and Rrp7 were lost, but the reduction in the helicase Rok1 was even more dramatic. Why was this ignored?

This finding has now been also mentioned in the text. Rok1 is a very transient and low abundant factor on the affinity-purified 90S particles (see iBAQ value), as compared to other major 90S factors. We plan to look into Rok1 and its possible interaction with Krr1-UTP-C in future studies.

6. In Fig 3d it looks like the total amount of Dhr1 is reduced in glucose but the ratio of Dhr1 to pre-ribosome in fractions 8-11 is not changed much between the two conditions. This seems counter to the authors' conclusion.

We have repeated this experiment using Flag-tagged Dhr1, which allowed us to probe directly for Dhr1 by Western blotting. This analysis revealed that under conditions of Dim2 expression (galactose), the major pool of Dhr1 is found as free protein on top of the sucrose gradient, while only a tiny amount of Dhr1 (hardly detectable by Western) is associated with fractions containing 80S ribosomes, which could be contaminants. Under conditions of Dim2 repression (glucose), much less free Dhr1 is found on top of the gradient, and more became associated with typical 90S particles, as visualized by Coomassie staining and Western probing (revised Fig. 3d)

7. In general, the Coomassie stained gels of pull downs appear to show heavy contamination with low molecular weight proteins. Are these mature ribosomal protein contaminants? No negative controls (untagged) are shown for the IPs.

We apologize for not labeling these proteins. These low molecular weight bands are ribosomal proteins, which on the hand are specific due to their association with 90S particles, but to a certain extent could be also unspecific due to contaminating ribosomes. We have repeated all these purifications under more stringent washing conditions to reduce these eventual contaminants.

As requested, we also show a mock control (revised Supp Fig. 2a), demonstrating that our split-tag affinity purification without a tagged protein is very clean.

8. The authors should explain how they conclude that Nob1 dissociates in the absence of Dim2 rather than is not recruited to begin with.

We have now better addressed the problem of Nob1-Dim2 recruitment. It is still possible that Dim2 when transferred on the platform site is substantially involved in Nob1 recruitment, but Dim2 depletion does not allow to address this.

Minor points:

8. In Fig 1, if depletion of Dim2 blocks 40S assembly, how is it that Rio2 binds pre-40S lacking Dim2? Does a population of particles slip through the assembly and export pathways lacking Dim2?

Not yet clear, but we have discussed redundant and separate targeting mechanisms.

And, no evidence is provided that the Coomassie-stained bands in the Rio2 pull down are related to pre-40S.

These bands have been identified by mass spectrometry and accordingly labeled in Fig. 1a.

9. Regarding the Nob1 in vitro cleavage, please clarify if cleavage was seen or not seen.

This part has been deleted in the revised manuscript, because it is less important for the overall story.

10. The authors mapped the interaction domains of Nob1 and Dim2 in Fig 1C but did not make use of this information to interpret structure or function.

This has been better addressed in the revised manuscript.

Reviewers' comments:

Reviewer #1 (Remarks to the Author):

In their revised version, the authors addressed some of the points brought up by the reviewers. The manuscript has been significantly improved, both in terms of data and text. As it stands, the most interesting results are those unveiling the association of Dim2 with Utp1/Utp14/Dhr1 within the 90S pre-ribosome. This finding, together with the information about Dim2-depleted particles, suggests that the protein is required for the Dhr1-mediated maturation step.

In addition to the Dim2-Dhr1 connection, the paper informs on two other separate issues: the role of Dim2 in Nob1 recruitment and the function of Krr1. In my view, these two parts of the study remain weak in terms of net contribution and, as mentioned in my previous review, make it disperse and deliver messages with little experimental support. Regarding the Dim2-Nob1 story, the authors nicely characterized the physical interaction between the two proteins, but no important role for such interaction was found. In regard to Krr1, presumed partners of the protein within the 90S particle were confirmed and a possible function in the docking of the UTP-C subcomplex was inferred, but just from results of one experiment. Despite these weaknesses, the manuscript now includes an important body of novel information and it will merit publication in Nature Communications if some of the defective aspects are improved. In particular, there are assessments not fully substantiated by data that have to be reformulated or backed by further experimental support.

The points to be addressed are the following:

1. In the first section of the results it is claimed that Dim2 has a dual role in 90S and pre-40S biogenesis. However, after reading the text, the two roles are unclear. One of them is in 90S particle maturation, but it is not precisely stated in the text. This can be easily fixed by incorporating a more explicit conclusion. Regarding the second role, the one in pre-40S particles, the description and conclusions are confusing and, in my view, not correct. The authors start by showing that Nob1 is absent from Rio2 particles in Dim2-depleted cells. In the current version there is no conclusion statement, but in the previous one it was reasoned that Nob1 was dissociated from pre-40S particles (this idea is now in the title of Figure 1 and in the legend of Figure 5).

As stated in my previous review, there is no evidence for claiming that Dim2 drives the recruitment of Nob1. Upon Dim2-depletion, no pre-40S particles are produced. There are aberrant 90S-like and 40S-like preribosomes that contain many early assembly factors (including

Krr1) (as seen in supplementary Figure 2B). The nature of the Rio2 complexes (in Dim2-depleted cells) shown in Fig. 1A is uncertain. Is Rio2 in the aberrant 90S-like and 40S-like particles? Is it in pre-assembly or post-assembly subcomplexes? To facilitate the interpretation of the results, the authors should analyze the presence of Rio2, Ltv1 and Nob1 in the 40S-like and 90S-like complexes in fractions 6 and 10 of the gradients shown in supplementary Figure 2B. If Rio2 and Ltv2, but not Nob1, are recruited to 40S-like particles in the absence of Dim2, it can be argued that the Dim2-mediated maturation step is required for Nob1 recruitment, but nothing else. Data by Woolls et al (JBC 2011) showed efficient recruitment of Nob1 to pre-40S complexes in the absence of Dim2, something that goes against the authors' current postulations. This should be commented or discussed. In the previous version of the manuscript, the Woolls' study was referenced as supportive of a role in Nob1 recruitment, but this was not correct.

Based on the above considerations, the first section of results requires additional data and a careful and clear argumentation of conclusions.

2. In the last section of the results it is concluded that Krr1 is required for the recruitment of the UTP-C module. This conclusion is based just on the compositional analyses of Utp10-FTpA-containing particles in Krr1-depleted cells. The authors should confirm these results with 90S particles purified using another bait (for example Utp1). Attention must be paid to possible changes in the association of Rrp5, a factor essential for the recruitment of both UTP-C and Rok1.

3. When proposing that Dim2 is required for Dhr1 activity, discuss that Dim2 has additional functions. Unlike Dim2, Utp14 and Dhr1 are not required for the A2 cleavage.

4. Statement in page 10: "Together, this data support a role of Dim2 in recruiting Dhr1 via the bridging factor Utp14 to the 90S particles, which in consequence could trigger progression in 90S-pre40S transition (see Discussion)".

Data indicate that Dim2 is required for Dhr1 activation, not recruitment.

5. Figure 5 legend. "The correct positioning of Dim2 at this site enables Dim2 to contact Utp14. Following this contact and further maturation steps, the helicase Dhr1 is eventually recruited to the particle through interaction with its co-activator Utp14"

Change this sentence. Dim2 is not required for Dhr1 recruitment.

6. Minor points.

- Page 9, typo: Utp14mulit-sup
- Page 10, error: instead of Fig. 4b should be Fig. 4a
- Supplementary figure 2. Panel of fraction 6 in bottom gradient. Some lines and letters are shifted and do not point to protein bands.

Reviewer #2 (Remarks to the Author):

Reviewer's comments, Sturm et al., NCOMMS-17-07747A

In their revised manuscript, Sturm and colleagues interrogated the role of Dim2 and Krr1 in 40S biogenesis and to establish the hierarchical and functional relationship between the two proteins as well as to other ribosome biogenesis factors (i.e. Nob1) during 90S and pre-40S assembly. In both the revised manuscript and reply to reviewers' comments the authors lay out their additional work in great detail. Not only have the authors addressed the concerns of all reviewers quite thoroughly, they have also carried out additional experiments including genetic analyses using Dim2 mutants, more thorough semi-quantitative mass spectrometry analysis, sucrose gradient centrifugation, in vitro binding assays for Nob1 and Dim2 interaction to support their hypothesis. The experiments are thorough and support their previous data. The authors have also expanded their manuscript with regards to previously published data and better related their findings to those works.

Overall, the authors have substantially improved the manuscript in regards to my but also the other reviewers' concerns, and I would therefore recommend the manuscript's acceptance for publication in Nature Communications.

Reviewer #3 (Remarks to the Author):

In this revision, the authors have added additional experimental work to address earlier concerns. The work is very clean and the results are very clear and clearly presented. However, I continue to have several reservations about the manuscript. Principally, the work is a collection of useful observations but could go further to address the mechanisms that drive 90S particle assembly beyond mapping protein contacts. As I noted in my first review "Combining the analysis of Dim2 and Krr1 in a single manuscript does add more information, but neither story is very strong and the focus is lost. The evidence for sequential assembly is modest." The authors mention several possible interesting aspects of 90S assembly but do not test them/

Additional comments:

The authors suggest that Krr1 may facilitate loading of Rps1 and/or Rps14. This should be tested. Especially because there is a clear interaction between Krr1 and Rps1 in the 90S structure and Utp22 interacts extensively with Rps1. A failure to load Rps1 in the absence of Krr1 would nicely explain the loss of Utp22 and Rrp7 upon Krr1 depletion.

The authors should take greater advantage of the recent 90S structures to discuss their work. In the 90S structure from Ye's group, there is no evident interaction between the resolved portions of Krr1 and Utp22. It is, of course possible, that unresolved extensions of these proteins are responsible for the observed interactions in vitro. However, this should be commented on.

The authors should discuss the previous work that leads to the conclusion that Dim2 relocates. This is mentioned at the end of the Introduction and in the Discussion. But, as it is, this point will be lost on the average reader who does not have a deep knowledge of 40S assembly.

Our point-to-point responses to the reviewers' comments

Reviewers' comments:

Reviewer #1 (Remarks to the Author):

In their revised version, the authors addressed some of the points brought up by the reviewers. The manuscript has been significantly improved, both in terms of data and text. As it stands, the most interesting results are those unveiling the association of Dim2 with Utp1/Utp14/Dhr1 within the 90S pre-ribosome. This finding, together with the information about Dim2-depleted particles, suggests that the protein is required for the Dhr1-mediated maturation step.

We are happy that that reviewer #1 now recommends publication of our manuscript in Nature Communications.

In addition to the Dim2-Dhr1 connection, the paper informs on two other separate issues: the role of Dim2 in Nob1 recruitment and the function of Krr1. In my view, these two parts of the study remain weak in terms of net contribution and, as mentioned in my previous review, make it disperse and deliver messages with little experimental support.

This study was performed with the goal to make a comparative functional and biochemical analysis of the structurally related KH domain proteins Dim2 and Krr1. We believe that combining these findings in one paper makes our story more complete.

Regarding the Dim2-Nob1 story, the authors nicely characterized the physical interaction between the two proteins, but no important role for such interaction was found. In regard to Krr1, presumed partners of the protein within the 90S particle were confirmed and a possible function in the docking of the UTP-C subcomplex was inferred, but just from results of one experiment. Despite these weaknesses, the manuscript now includes an important body of novel information and it will merit publication in Nature Communications if some of the defective aspects are improved. In particular, there are assessments not fully substantiated by data that have to be reformulated or backed by further experimental support.

The points to be addressed are the following:

1. In the first section of the results it is claimed that Dim2 has a dual role in 90S and pre-40S biogenesis. However, after reading the text, the two roles are unclear. One of them is in 90S particle maturation, but it is not precisely stated in the text. This can be easily fixed by incorporating a more explicit conclusion.

We have clearly stated that Dim2 has a dual role in 90S and pre-40S biogenesis within the introduction:

'To date, Dim2 has been predominantly studied in the context of the late pre-40S maturation pathway though its association with Nob1^{25, 28}, although a role in early ribosome assembly and A₂ cleavage has been reported³¹.'

as well as in the Discussion:

'After dismantling the 90S pre-ribosome and liberation of the pre-40S moiety, Dim2 continues its job as a biogenesis factor in the subsequent maturation steps, in which it functions at the platform/neck/head region of the derived pre-40S particle.'

We include now a short conclusion remark within the result section:

'When Enp1 was affinity-purified from Dim2-depleted cells, the typical profile of the Enp1 co-enriched bands, which are mainly late pre-40S factors (e.g. Rrp12, Tsr1, Nob1, Dim1, Dim2), changed in favor of a massive co-enrichment of normally underrepresented early 90S factors including UTP-A, UTP-B, UTB-C and U3 snoRNP subunits (Fig. 1a). This finding underscores Dim2's essential role of in 90S ribosome biogenesis.

Regarding the second role, the one in pre-40S particles, the description and conclusions are confusing and, in my view, not correct. The authors start by showing that Nob1 is absent from Rio2 particles in Dim2-depleted cells. In the current version there is no conclusion statement, but in the previous one it was reasoned that Nob1 was dissociated from pre-40S particles (this idea is now in the title of Figure 1 and in the legend of Figure 5).

Such a conclusion statement has now been added, also in the context of our new data, which revealed that only Nob1, but not Tsr1 and Rio2 are absent from altered pre-40S particles, isolated from Dim2-depleted cells (revised Supplementary Fig. 2c)

As stated in my previous review, there is no evidence for claiming that Dim2 drives the recruitment of Nob1. Upon Dim2-depletion, no pre-40S particles are produced. There are aberrant 90S-like and 40S-like preribosomes that contain many early assembly factors (including Krr1) (as seen in supplementary Figure 2B). The nature of the Rio2 complexes (in Dim2-depleted cells) shown in Fig. 1A is uncertain. Is Rio2 in the aberrant 90S-like and 40S-like particles? Is it in pre-assembly or post-assembly subcomplexes? To facilitate the interpretation of the results, the authors should analyze the presence of Rio2, Ltv1 and Nob1 in the 40S-like and 90S-like complexes in fractions 6 and 10 of the gradients shown in supplementary Figure 2B.

If Rio2 and Ltv2, but not Nob1, are recruited to 40S-like particles in the absence of Dim2, it can be argued that the Dim2-mediated maturation step is required for Nob1 recruitment, but nothing else.

We have analyzed the presence of Rio2, Trs1, Nob1 and Dim2 in the 40S-like fraction #6 by Western blotting. This additional data is shown in the revised Supplementary Fig. 2c. This clearly revealed that Nob1 is strongly diminished as compared to the other 40S biogenesis factors Rio2 and Trs1, suggesting that Dim2 depletion causes a rather specific defect in Nob1 recruitment. However, we carefully interpret this result as suggested by this reviewer by saying: 'Consistent with these findings, Western blot analysis revealed reduced Nob1 levels in comparison to the other pre-40S assembly factors Trs1 and Rio2 (Supplementary Fig. 2d), indicating that a Dim2-mediated maturation step is required for Nob1 recruitment.'

Data by Woolls et al (JBC 2011) showed efficient recruitment of Nob1 to pre-40S complexes in the absence of Dim2, something that goes against the authors' current postulations. This should be commented or discussed. In the previous version of the manuscript, the Woolls' study was referenced as supportive of a role in Nob1 recruitment, but this was not correct.

We have now discussed the Woolls paper (JBC 2011) in comparison to our findings in the Results section. However, we find that our data and those from the Woolls study are highly similar. We show that Nob1 levels are reduced in tandem-affinity purified pre-ribosomal particles upon Dim2 depletion; however, Nob1 associates normally with pre-ribosomes when the interaction between Dim2 and Nob1 is altered. The Woolls et al. study (JBC 2011) shows that Nob1 is found in large quantities in the upper part of the sucrose gradient (free pool) upon Dim2 depletion, suggesting a recruitment defect of Nob1 to the pre-40S particles upon Dim2 depletion. Furthermore, by using *Dim2* mutants incapable of binding Nob1, Nob1 associates normally with pre-ribosomes (Woolls et al. 2011, Figure 6).

2. In the last section of the results it is concluded that Krr1 is required for the recruitment of the UTP-C module. This conclusion is based just on the compositional analyses of Utp10-FTpA-containing particles in Krr1-depleted cells. The authors should confirm these results with 90S particles purified using another bait (for example Utp1). Attention must be paid to possible changes in the association of Rrp5, a factor essential for the recruitment of both UTP-C and Rok1.

We have performed this suggested experiment using another bait Utp1. Importantly, we have confirmed with this bait that Utp22 recruitment to 90S particles is defective upon Krr1 depletion (revised Fig. 5a). Regarding Rrp5, this biogenesis factor is clearly visible by Coomassie staining in our affinity purifications when Krr1 is depleted (Fig. 5a), but was not drastically changed. This has been mentioned in the text. Semi-quantitative mass spectrometry analysis (see Extended Excel file) confirms this observation. This data are now mentioned in the revised manuscript.

3. When proposing that Dim2 is required for Dhr1 activity, discuss that Dim2 has additional functions. Unlike Dim2, Utp14 and Dhr1 are not required for the A2 cleavage.

According to our interpretation of the published literature, Utp14 and Dhr1 were shown to be required for A2 cleavage (Sardana et al. 2014, Sardana et al. 2013).

4. Statement in page 10: "Together, this data support a role of Dim2 in recruiting Dhr1 via the bridging factor Utp14 to the 90S particles, which in consequence could trigger progression in 90S-pre40S transition (see Discussion)".

Data indicate that Dim2 is required for Dhr1 activation, not recruitment.

We apologize for this mistake. We now say: "Together, this data support a role of Dim2 in activating Dhr1 via the bridging factor Utp14, which in consequence could trigger progression in 90S-pre40S transition (see Discussion)".

5. Figure 5 legend. "The correct positioning of Dim2 at this site enables Dim2 to contact Utp14. Following this contact and further maturation steps, the helicase Dhr1 is eventually recruited to the particle through interaction with its co-activator Utp14"

Change this sentence. Dim2 is not required for Dhr1 recruitment.

Here, we did not mean that Dim2 is directly required for the Dhr1 recruitment, but we can see that the phrasing is misleading and have therefore corrected this: "The correct positioning of Dim2 at this site allows the subsequent activation of Dhr1 by its activator Utp14."

6. Minor points.

- Page 9, typo: Utp14mlit-sup
corrected

- Page 10, error: instead of Fig. 4b should be Fig. 4a
corrected

- Supplementary figure 2. Panel of fraction 6 in bottom gradient. Some lines and letters are shifted and do not point to protein bands.
corrected

Reviewer #2 (Remarks to the Author):

Reviewer's comments, Sturm et al., NCOMMS-17-07747A

In their revised manuscript, Sturm and colleagues interrogated the role of Dim2 and Krr1 in 40S biogenesis and to establish the hierarchical and functional relationship between the two proteins as well as to other ribosome biogenesis factors (i.e. Nob1) during 90S and pre-40S assembly. In both the revised manuscript and reply to reviewers' comments the authors lay out their additional work in great detail. Not only have the authors addressed the concerns of all reviewers quite thoroughly, they have also carried out additional experiments including genetic analyses using Dim2 mutants, more thorough semi-quantitative mass spectrometry analysis, sucrose gradient centrifugation, in vitro binding assays for Nob1 and Dim2 interaction to support their hypothesis. The experiments are thorough and support their previous data. The authors have also expanded their manuscript with regards to previously published data and better related their findings to those works.

Overall, the authors have substantially improved the manuscript in regards to my but also the other reviewers' concerns, and I would therefore recommend the manuscript's acceptance for publication in Nature Communications.

Reviewer #3 (Remarks to the Author):

In this revision, the authors have added additional experimental work to address earlier concerns. The work is very clean and the results are very clear and clearly presented. However, I continue to have several reservations about the manuscript. Principally, the work is a collection of useful observations but could go further to address the mechanisms that drive 90S particle assembly beyond mapping protein contacts.

This study was conducted with the goal to perform a comparative functional and biochemical analysis of the structurally related KH domain proteins Dim2 and Krr1, which revealed for both proteins several interesting findings, which addressed the mechanisms that drive 90S particle assembly and further maturation towards pre-40S particles. We believe that these combined data, which not only mapped protein contacts, make our story more complete.

As I noted in my first review “Combining the analysis of Dim2 and Krr1 in a single manuscript does add more information, but neither story is very strong and the focus is lost. The evidence for sequential assembly is modest.” The authors mention several possible interesting aspects of 90S assembly but do not test them/

We have included additional data concerning the role of Krr1 in Rps1 and Rps14 recruitment (see below). We were able to show that Krr1 clearly contributes in preparing the later binding site of Dim2 by recruiting both ribosomal proteins to the platform domain of the later pre-40S particle. Thus, we could show a direct link between these two structurally related assembly factors.

Additional comments:

The authors suggest that Krr1 may facilitate loading of Rps1 and/or Rps14. This should be tested. Especially because there is a clear interaction between Krr1 and Rps1 in the 90S structure and Utp22 interacts extensively with Rps1. A failure to load Rps1 in the absence of Krr1 would nicely explain the loss of Utp22 and Rrp7 upon Krr1 depletion.

We have now included Western blots against Rps14, Rps5 and Rps8 to show that Rps14 levels are reduced upon Krr1 depletion, while Rps5 and Rps8 levels are not affected. In addition, we performed mass-spectrometry of the Coomassie stained Rps1 and Rps5 bands (Fig. 5a) and found that Rps1 is strongly reduced in comparison to the nearby Rps5 in Krr1-depleted cells. This data are consistent with a role of Krr1 in recruitment of Rps14-Rps1 to the pre-40S platform area within the 90S, where also Utp22-Rrp7 are located. This new data are discussed in the context of the current 90S cryo-EM structures (see Discussion).

The authors should take greater advantage of the recent 90S structures to discuss their work. In the 90S structure from Ye's group, there is no evident interaction between the resolved portions of Krr1 and Utp22. It is, of course possible, that unresolved extensions of these proteins are responsible for the observed interactions in vitro. However, this should be commented on. The authors should discuss the previous work that leads to the conclusion that Dim2 relocates. This is mentioned at the end of the Introduction and in the Discussion. But, as it is, this point will be lost on the average reader who does not have a deep knowledge of 40S assembly.

We have now commented on this in the discussion, and also modified our model (Fig. 6).

“Our data also allow us to speculate how another key module of the 90S pre-

ribosome, the UTP-C complex members Utp22 and Rrp7, could be recruited to the 90S pre-ribosome via the help of Krr1. Interestingly, the recently published cryo-EM map of the *Saccharomyces cerevisiae* 90S particle revealed that Utp22 and Rrp7 are located in close proximity to Krr1 and Rps1-Rps14 without a direct contact between Krr1 and Utp22-Rrp7⁹. However, not all parts of Krr1 and the UTP-C module could be modeled into the 90S structure. Based on our findings that Krr1-Utp22-Rrp7 form a robust complex, it is conceivable to assume that Krr1 interacts directly with Utp22, Rrp7 or both within the context of the 90S pre-ribosome. Taken all this data together, the location of Krr1 at the evolving head-platform area of the 90S pre-ribosome explains well how this KH domain protein together with neighboring ribosomal proteins such as Rps14 and Rps1 could play a role for recruitment of the UTP-C complex to this site of the 90S pre-ribosome.”

Reviewers' Comments:

Reviewer #1 (Remarks to the Author):

The revised version of the manuscript addresses the points I had raised in my last review. The work improves our molecular understanding of some crucial processes behind ribosome assembly. I recommend publication in Nature Communications.

Reviewer #3 (Remarks to the Author):

In this revision, the authors have now provided evidence that Krr1 promotes the association of Rps1 and Rps14 and rationalize the loss of Utp22, explained by recent high resolution cryo-EM structures. The authors have satisfied my concerns and I can now recommend the manuscript for publication.